# Identification of distinct functional thymic programming of fetal and pediatric human γδ thymocytes via single-cell analysis

Guillem Sanchez Sanchez [1,2,3], Maria Papadopoulou [1,2,3], Abdulkader Azouz [2,3], Yohannes Tafesse [1,2,3], Archita Mishra[4,5], Jerry K. Y. Chan[6,7,8], Yiping Fan[6,7,8], Isoline Verdebout [1,2,3], Silvana Porco[1,2,3], Frédérick Libert[9], Florent Ginhoux [4], Bart Vandekerckhove[10], Stanislas Goriely [2,3] & David Vermijlen [1,2,3,11] ✉

Developmental thymic waves of innate-like and adaptive-like γδ T cells have been described, but the current understanding of γδ T cell development is mainly limited to mouse models. Here, we combine single cell (sc) RNA gene expression and sc γδ T cell receptor (TCR) sequencing on fetal and pediatric γδ thymocytes in order to understand the ontogeny of human γδ T cells. Mature fetal γδ thymocytes (both the Vγ9Vδ2 and nonVγ9Vδ2 subsets) are committed to either a type 1, a type 3 or a type 2-like effector fate displaying a wave-like pattern depending on gestation age, and are enriched for public CDR3 features upon maturation. Strikingly, these effector modules express different CDR3 sequences and follow distinct developmental trajectories. In contrast, the pediatric thymus generates only a small effector subset that is highly biased towards Vγ9Vδ2 TCR usage and shows a mixed type 1/type 3 effector profile. Thus, our combined dataset of gene expression and detailed TCR information at the single-cell level identifies distinct functional thymic programming of γδ T cell immunity in human.

γδ T cells are the 'third' type of lymphocytes, besides αβ T cells and B cells, that can rearrange gene segments at the DNA level in order to generate variable antigen receptors[1,2]. These three cell lineages have been conserved seemingly since the emergence of jawed vertebrates, with the notable exception of squamate reptiles[3], while a similar tripartite subdivision exists even in jawless vertebrates such as lamprey and hagfish[4,5]. Emerging evidence suggests that their role in early life immunity might be a critical factor for this striking evolutionary conservation[2,6–15]. Indeed, human γδ T cells have been shown to react vigorously to infections in utero[10,16] and early environmental post-natal exposure[14,15] and in mouse models γδ T cells confer protection against parasite and viral infections in early life and/or when the αβ T cell compartment is compromised[6,17,18]. Furthermore, besides protection against infection, mouse models indicate that fetal-derived γδ T cells may play crucial physiological roles such as thermoregulation and the development of brain/short-term memory[19].

[1]Department of Pharmacotherapy and Pharmaceutics, Université Libre de Bruxelles (ULB), Brussels, Belgium. [2]Institute for Medical Immunology, Université Libre de Bruxelles (ULB), Gosselies, Belgium. [3]ULB Center for Research in Immunology (U-CRI), Université Libre de Bruxelles (ULB), Brussels, Belgium. [4]Singapore Immunology Network (SIgN), A*STAR, Singapore 138648, Singapore. [5]Telethon Kids Institute, University of Western Australia, Perth, Australia. [6]Department of Reproductive Medicine, KK Women's and Children's Hospital, Singapore 229899, Singapore. [7]Experimental Fetal Medicine Group, Yong Loo Lin School of Medicine, National University of Singapore, Singapore 117597, Singapore. [8]Academic Clinical Program in Obstetrics and Gynaecology, Duke-NUS Medical School, Singapore 229899, Singapore. [9]BRIGHTcore ULB-VUB, Université Libre de Bruxelles (ULB), Brussels, Belgium. [10]Department of Diagnostic Sciences, Ghent University, Ghent, Belgium. [11]Walloon Excellence in Life Sciences and Biotechnology (WELBIO), Wavre, Belgium. ✉e-mail: David.Vermijlen@ulb.be

Translation of γδ T cell biology findings from mouse models toward human are complicated by the lack of conservation of the γ and δ loci[20,21]. For example, in contrast to the conservation of αβ TCR-expressing innate T cells (MR1/metabolite-reactive MAIT, CD1d/lipid-reactive iNKT), human phosphoantigen-reactive Vγ9Vδ2 T cells do not exist in mice, and, vice versa, no human homolog of mouse dendritic epidermal T cells (DETC, γδ T cells highly enriched in the mouse skin epidermis) has been identified[5]. In human, it is becoming increasingly clear that the phosphoantigen-reactive γδ T cells are innate-like T cells, while nonVγ9Vδ2 T cells adopt an adaptive nature[2,22–24]. Despite this increasing knowledge about the effector functions and TCR recognition modalities, only little is known about the thymic development of human γδ T cells.

Like αβ T cells, γδ T cells are generated in the thymus where rearrangement of V, D, and J gene segments takes place in order to form a TCR at their cell surface. Conventional CD4 and CD8 αβ T cells leave the thymus as naïve T cells that can develop into the right functional effector cells in the periphery, depending on the type of pathogen encounter, such as cytotoxic CD8 αβ T cells and type 1, 2 or type 3 CD4 αβ T helper cells. Recent single-cell analysis revealed, however, that this CD4+ effector T cell pool generated in response to various pathogens cannot be easily parsed into discrete T helper lineages but instead forms a continuum of polarized phenotypes that is shaped by the specific pathogens[25,26]. We have recently shown that human fetal γδ thymocytes are already functionally programmed and are highly enriched for several 'human-specific' invariant/public TCR sequences[27,28]. Whether human γδ T cells are pre-committed towards distinct functional effector programs and whether this is linked to the expression of specific invariant/public fetal thymic γδ TCR sequences is not known. In particular, the distinct features of Vγ9Vδ2 T cells suggest that they could follow different rules during their thymic development.

Here, we took advantage of combining single cell (sc) RNA gene expression (RNA-seq) with sc γδ TCR sequencing to unravel human γδ T cell development. As such we identify developmental stage-specific thymocyte effector clusters and their concomitant TCR repertoire and differentiation pathways.

## Results

### Experimental design

In order to obtain insight into the effector programming in the human fetal thymus, we performed sc RNA/TCR sequencing on γδ thymocytes from six fetal thymuses, in parallel with three pediatric thymuses (Fig. 1A), followed by confirmation of selected findings at protein level by flow cytometry. We sorted γδTCR+CD3+ thymocytes (Supplementary Fig. 1A) before applying the scRNA/TCR sequencing protocol in order to link unequivocally particular gene expression profiles that can be shared by other (innate-like) lymphocytes[29–32]. Flow cytometry results showed a negative correlation between the gestation age of the fetus and the frequency of γδ thymocytes (Fig. 1B) and of Vγ9Vδ2 T cells among γδ thymocytes (Fig. 1C). From this analysis we selected for the scRNA/TCR experiments a series of fetal thymuses ranging from 14 to 22 weeks of gestation time, allowing the analysis of γδ thymocytes along these different ages, in particular the comparison of Vγ9Vδ2 and nonVγ9Vδ2 thymocyte development. The human pediatric γδ thymocytes, possessing only a low percentage of Vγ9Vδ2 T cells[27] (Fig. 1C), did not show a correlation with post-natal age and we selected the ages 4.0, 4.5, and 11.0 years (Supplementary Fig. 1B).

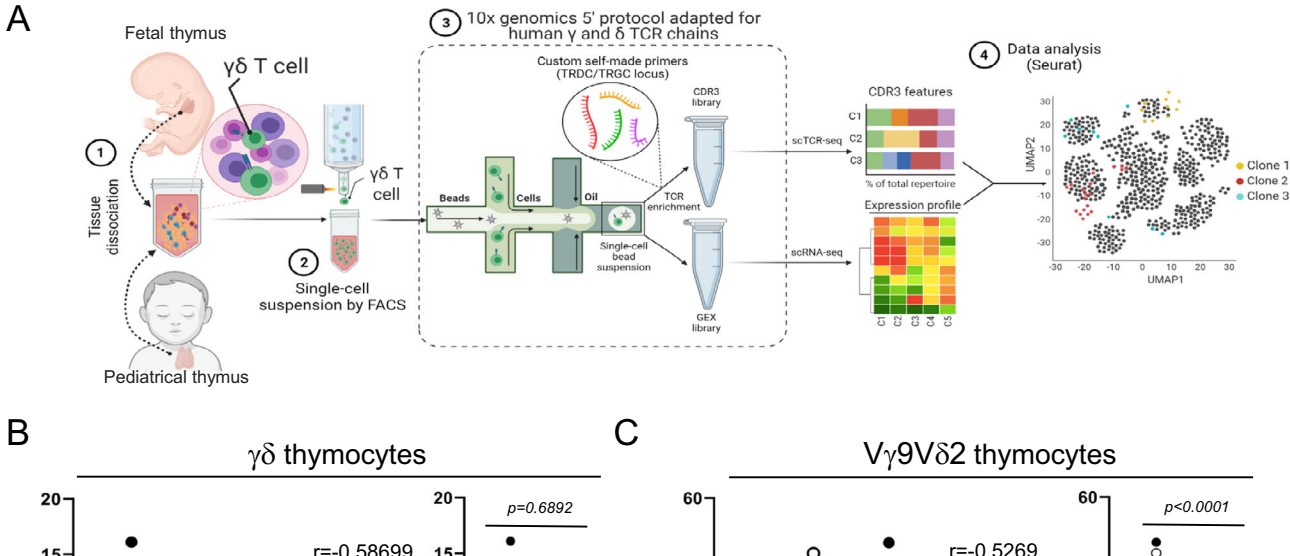

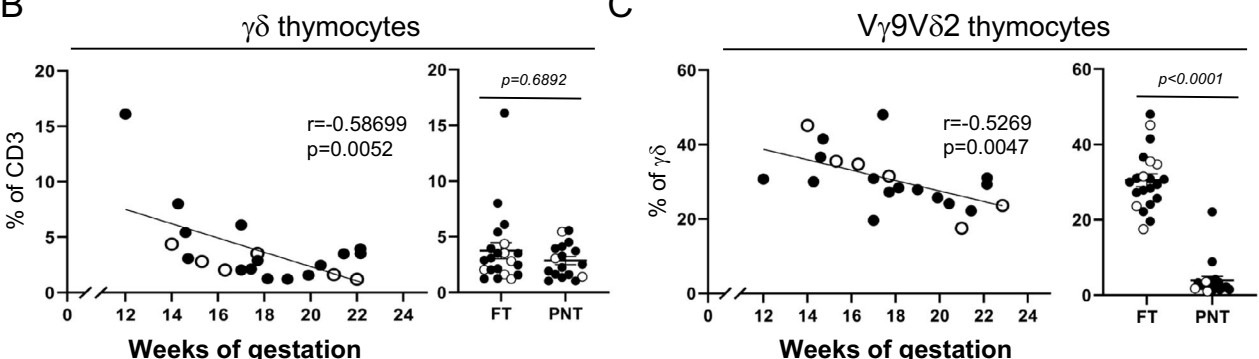

**Fig. 1 | Experimental approach. A** γδ T cells were sorted from human fetal and pediatric thymuses and were subjected to a modified version of 10x genomics 5′ protocol in order to amplify CDR3δ and CDR3γ regions. Flow chart created with BioRender.com. **B** Frequency (%) of γδ thymocytes out of total CD3+ living cells. **C** Frequency (%) of Vγ9Vδ2 thymocytes out of γδ+ thymocytes. R and *p* values (two-tailed) in **B**, **C** were obtained by doing Spearman correlation test in the gestation age graphs, while dot plots were analyzed by two-tailed unpaired *t*-test. **B**, **C** White dots indicate samples used in the sc RNA/TCR-seq experiments. "FT" group: fetal thymus. "PNT" group: post-natal thymus/pediatric thymus. Source data are provided as a Source Data file. See also Supplementary Fig. 1.

## scRNA sequencing identifies heterogeneous immature and mature γδ thymocyte clusters in the human fetal thymus

After quality control and integration of the fetal γδ thymocyte datasets (n = 6), a total of 16,508 γδ thymocytes were retained for downstream analysis (average gene number = 1875; average Unique Molecular Identifier or UMI = 2851) (Supplementary Fig. 2A). Plotting the cells by Uniform Manifold Approximation and Projection (UMAP), led to the identification of 11 distinct clusters with some of them being more enriched in certain subjects (Fig. 2A, Supplementary Fig. 2B). Assessment of thymocyte maturation markers[33–36] displayed a broad landscape of distinct maturing cell states where the most immature ones were found in clusters 2 (c2), c7, c8, and c10 (Fig. 2A, B). We identified several functional states based on the most differentially expressed genes (DEGs) from the different clusters (Fig. 2C, Supplementary Data 1). C8, c9 and c10 were enriched for cell-cycle genes like *TYMS*, *MKI67*, or *TOP2A* (Fig. 2C), and, supported by "Cell Cycle Score" (Supplementary Fig. 2C), were classified as cycling clusters. C2 and c7 shared a transcriptomic profile enriched for immature thymocyte markers (Fig. 2C), with c2 displaying an enrichment for the mouse γδ T lineage-specific TF *TOX2*[34], while c7 showed increased expression of the adhesion molecule *SELL* (Fig. 2C). C2 shared part of its transcriptomic profile with cells from c5 including genes such *PDCD1* (PD1) (these clusters are plotted next to each other, Fig. 2A), with c5 enriched as well for other genes induced as a consequence of TCR signaling such as *ACTN1*[37], *P2RX5*[38], or *XCL1*[39], and other markers that characterize T cell subsets that undergo agonistic TCR signaling including *IKZF2*[40,41] and *MME*[31,42] (Fig. 2C, Supplementary Fig. 14, Supplementary Data 1). The expression of the chemokine-genes *XCL1* and *XCL2* in c5 was only shared with c4, with this last one being also enriched for other chemokines like *CCL4* and *CCL5* (Fig. 2C, Supplementary Data 1). C4 was also defined by transcripts associated with a cytotoxic profile like granzymes (Fig. 2C, Supplementary Fig. 14, Supplementary Data 1). Cluster 3 was enriched for cell membrane receptors *CCR6*, *IL7R*, and *KLRB1* (CD161) (Fig. 2C; Supplementary Data 1). Cluster 6 expressed genes with diverse functionalities such as *CTSL*, *ANXA1*, or *CD40LG* co-stimulation molecule (Fig. 2C). Cells from c11 displayed a unique signature featuring several type I IFN-inducible genes like *ISG15*, *MX1*, or *IFI6*, a result that might be connected to previous reports describing tonic type I IFN signaling as crucial for the development of mouse αβ thymocytes[43]. Of note, the immature (Fig. 2B) and cycling (Supplementary Fig. 2C) c10 expressed high levels of *CD4*, *CD8*, *RAG1*, and *PTCRA*, as well as members involved in the Notch signaling pathway (*NOTCH3* and *HES4*) (Supplementary Fig. 2D left panel); thus, this cluster likely corresponds to the CD4+CD8+ double positive γδ thymocyte population that has been identified in the human thymus that is Notch-dependent and rearranges the TRA locus[44]. Interestingly, flow cytometry experiments showed increased proportions of immature thymocytes (defined by CD1a expression[44]) with increasing gestation time exclusively in the γδ lineage (Supplementary Fig. 2E) and also confirmed the presence of the small double positive immature cell subset (CD1a+CD4+CD8+) identified by sc transcriptomics (Supplementary Fig. 2D right panel).

In summary, sc RNA-seq of fetal γδ thymocytes revealed a heterogenous set of immature and mature cell clusters.

## Mature fetal γδ thymocytes are committed to either a type 1, a type 3, or a type 2-like effector fate

Next, we aimed to obtain more insight into the most mature clusters c3, c4, and c6 identified in the fetal thymus (Fig. 2B) that appear to express DEG linked to effector profiles (Fig. 2C, Supplementary Data 1). Since thymocytes usually leave the thymus after maturation, we computed a score to assess the thymic egress potential of the mature γδ T cells in our dataset (Supplementary Data 5). This analysis indicated that c3, c4, and c6 had the highest thymic egress score among all clusters present in the fetal thymus (Fig. 3A, left panel), including the expression of the thymic egress master regulator *KLF2* (Supplementary

Fig. 3A)[45]. These mature clusters also shared expression of genes associated with a mature profile such as *CD44*, *CD69*, *GIMAP4*, and *CD55*, genes associated with inflammation such as *TNF* or *FASLG* and genes associated with immune regulation such as *ZFP36* (encoding Tristetraprolin, TTP) and *KLRB1* (CD161)[35,46–50] (Supplementary Fig. 3A, Supplementary Data 2).

After assessing their common egress potential and common gene expression, we analyzed in depth the differences in the transcriptional profile between c3, c4, and c6 in order to gain insight in their possible roles in the periphery (Fig. 3A–C). Cluster 4 was associated to a type 1 immunity profile based on its high expression of *IFNG*, *TBX21* (Tbet), *IL12RB2*, and *CXCR3* transcripts, as well as cytotoxicity-related genes such as *EOMES*, *GNLY*, *PRF1* (perforin), *TYROBP* (DAP12), *GZMA*, *GZMK*, *NKG7* (resulting in a high Type 1 score, Fig. 3A, Supplementary Data 5) and natural killer (NK) cell receptors *KLRC1* (NKG2A/B), *KLRC3* (NKG2E) and *KLRD1* (CD94) (Fig. 3B–C; Supplementary Fig. 14; Supplementary Data 3). These cells also expressed the mouse type 1 γδ markers *CD27* and *IL2RB* (CD122)[51–53] (Supplementary Fig. 14; Supplementary Data 3). In contrast to c4, c3 cells were associated with a type 3 immunity effector function based on two observations in this cluster: (i) a similar transcriptomic profile as mouse type 3 γδ cells (*RORC, CCR6, IL23R, IL7R, MAF, AHR, CLEC7A* (Dectin-1), *RORA, SLAMF1, BLK, JAML* (Amica1), or *CXCR6*) resulting in a high 'Type 3 score' (Fig. 3A, B; Supplemental Tables 1C and 2; Supplementary Fig. 14)[51,54–60]; (ii) the highly specific expression of *IL17A* and *IL17F* transcripts (Fig. 3C). Other genes present in this cluster included the ILC3 markers *KIT* and *NCR3*, as well as soluble factors such as *FLT3LG* and *AQP3* transporter (Fig. 3B; supplemental Table 1C), the last one being recently linked with IL17A production in lung γδ T cell upon influenza[61]. Finally, c6 expressed *IL4R, CD40LG, PLAC8, PLAUR, CCR4*, and *CD4* which can be linked to a type 2 immune profile[62–65] while the enriched expression of *IL2, ICOS, CD28, IL6ST, LEF1, TCF7*, and *STAT3* are features of a non-mutually exclusive T follicular helper-like function[66–68] (Fig. 3B; Supplementary Data 3; Supplementary Fig. 14). As the type 2 signature cytokines *IL4, IL5*, and *IL13* were not or only poorly expressed (Fig. 3C, Supplementary Fig. 14), this cluster is rather 'type 2-like' but for simplicity we will use the term 'type 2' from now on. Interestingly, assessment of *ZBTB16* (PLZF) expression levels in the 3 effector clusters showed a similar picture as described for mouse iNKT thymic cell subsets, with the highest levels of expression in the putative type 2 cluster[69–72] (Fig. 3D). We further validated the annotation of effector clusters by Gene Ontology (GO) enrichment analysis, linking c4 to type 1 effector gene sets such as viral response, NK cell activity or TCR associated signaling pathways, while c3 was associated to Th17 gene sets and brown fat cell differentiation, and c6 to regulation of B cell and humoral immune responses (Supplementary Fig. 3B).

In the mouse model, fetal thymic functional waves of γδ T cell production have been described[73]. Therefore, we analyzed whether there was an association between gestation age and the level and type of effector programming among the γδ thymocytes in the human fetus. The earlier the gestation time, the higher the percentage was of effector γδ thymocytes (Fig. 3E, top panel). Furthermore, among the effector γδ thymocytes, the type 3 program was highly abundant in 14 wk fetuses and declined from then onwards (till 22 wk of gestation), while there was a concomitant increase of the type 1 module; the type 2 program remained relatively stable (Fig. 3, bottom panel). Furthermore, the 'strength' of the type 3 program within the type 3 cluster was highest at the earliest gestation time (Supplementary Fig. 4A). The high prevalence of this type 3 programming (abundance and strength) at the earliest gestation times is possibly due to the abundance of a putative type 3 precursor[74–76] (Supplementary Fig. 4B).

Finally, in order to validate the division into type 1, 3, and 2 effector programs at the protein level, we selected membrane markers from the scRNA data of the three effector clusters in order to be verified by flow cytometry: CD94, CD8a, and NKG2D (CD314) for type 1, CD161, CD26, and CCR6 (CD196) for type 3 and CD4, CCR4 (CD194),

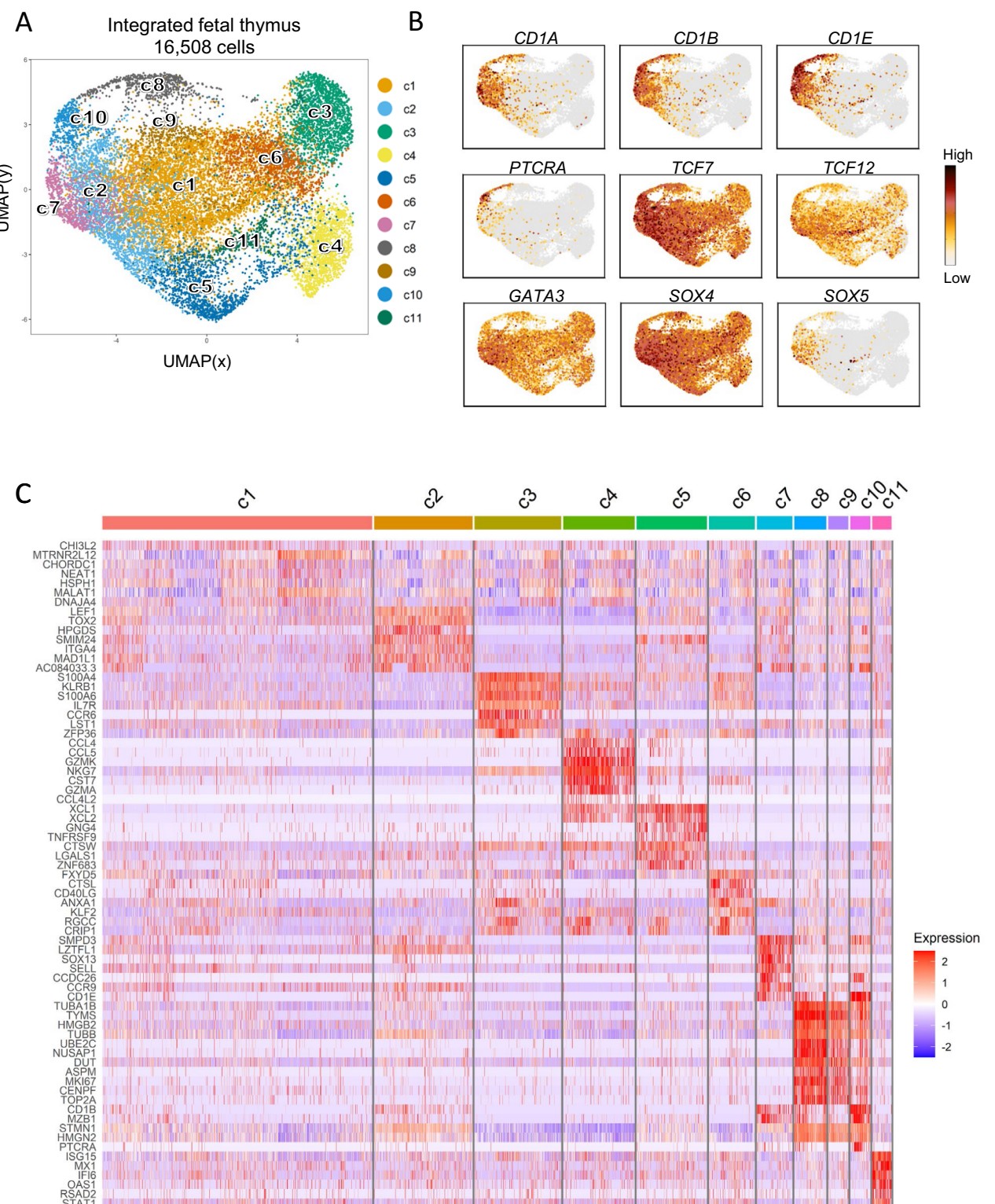

**Fig. 2 | scRNA sequencing identifies heterogenous immature and mature γδ thymocyte clusters in the human fetal thymus.** Single γδ thymocytes from six different fetal subjects were sorted in 3 independent experiments. After quality control, 16,508 cells were subjected to downstream analysis. **A** UMAP visualization of the annotated clusters obtained after integration of the six samples. **B** Expression levels of certain genes reported to change during thymocyte maturation. **C** Heatmap showing row-scaled Log-2 fold change (logFC) expression values of differentially expressed genes (DEGs). Each fine column is a cell. List of DEGs is available in Supplementary Data 1. See also Supplementary Fig. 2.

and ICOS (CD278) for type 2. This analysis confirmed the pattern of expression of these 'type 1/3/2' markers among fetal γδ thymocytes at the protein level (Fig. 3F; Supplementary Fig. 5).

Thus, mature fetal γδ thymocytes are directed to three distinct effector fates related to either type 1, type 3, or type 2-like immunity in a wave-like pattern depending on gestation age.

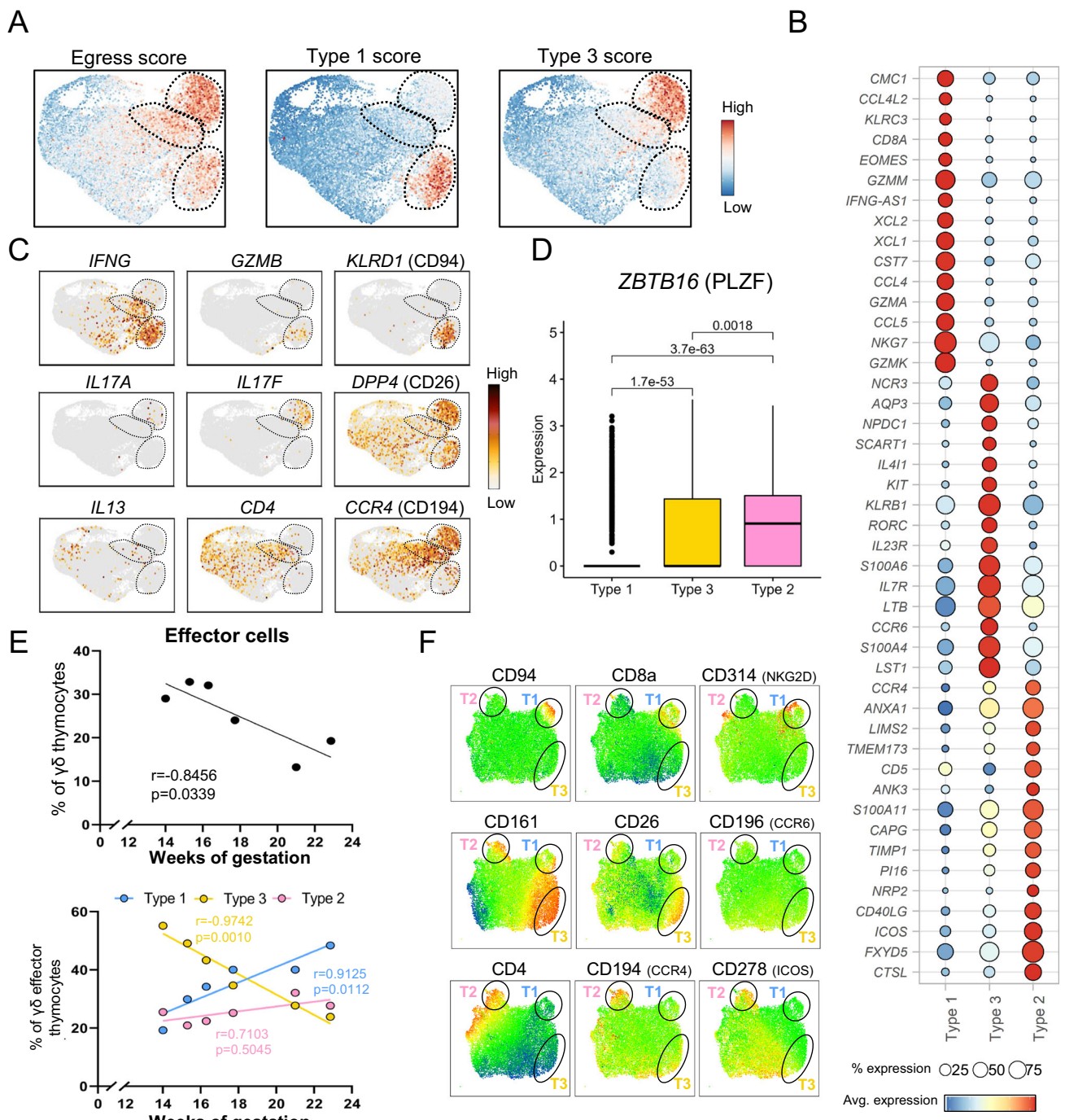

**Fig. 3 | Mature fetal γδ thymocytes are committed to either a type 1, a type 3, or type 2-like effector fate. A** Projection of the egress, Type 3 (γδ17) and Type 1 (cytotoxicity) scores computed with distinct sets of genes (Supplementary Data 5); each cell is colored based on its individual score. **B** Dot plot heatmap displaying row-scaled Log2-fold change (logFC) expression values of selected DEGs in the comparison between the three distinct effector clusters. Each dot represents the average expression profile of all the cells in the effector cluster. **C** UMAP plots displaying expression levels of selected genes that allow a phenotypic characterization of the 3 effector clusters (type 1, type 3, and type 2-like clusters). **D** Box plot comparing PLZF expression levels in the distinct effector clusters; *P* values (two-tailed) were calculated by Wilcoxon test with Benjamin–Hochberg correction (type 1: *n* = 1461 cells, type 3: *n* = 1867 cells and

type 2-like: *n* = 1054 cells). The three horizontal lines of the box–whisker plot represent the higher quartile, median, and lower quartile, respectively. The whiskers stretch from each quartile to the maximum or minimum. **E**, top panel: Abundance of sc effector cells across gestation time. **E**, bottom panel: Abundance of distinct sc effector cells (type 1, type 3, and type 2-like) across gestation time. R and p values (two-tailed) were determined by Spearman correlation. **F** UMAP plots derived from flow cytometry data indicating the expression levels of distinct surface markers; UMAP plots include 17,091 live CD3⁺ γδ⁺ thymocytes from 4 different thymuses; black circles are used to facilitate the location of the effector clusters: T1 = Type 1, T3 = Type 3, T2 = Type 2-like. Source data are provided as a Source data file. See also Supplementary Figs. 3–5.

## Type 1, type 3, and type 2 γδ thymocytes are enriched for public CDR3 features upon maturation

We have previously shown that the fetal γδ thymocyte TCR repertoire, compared to the post-natal counterpart, is highly enriched for the usage of particular *TRDV* and *TRGV* gene segments (*TRDV2, TRGV8, TRGV9*), low N additions, and public CDR3 sequences[27,28]. Here, by combining RNA-seq and γδ TCR data at the single-cell level, we assessed whether these features are influenced by the maturation or effector states. Our sc γδ TCR protocol identified a CDR3δ and a CDR3γ chain in 68 and 52% of the cells, respectively (mean values from the 6 fetuses). We detected a low percentage, around 2.3%, of γδ thymocytes expressing double (productive) CDR3δ sequences; comparable frequencies of double expressors have been detected in human peripheral γδ T cells[77]. The fetal γδ thymocytes were in general enriched for *TRDV2* usage both in immature/maturing and effector clusters, but the highest *TRDV2* frequency was found in the effector clusters (Fig. 4A). Consequently, the proportion of other *TRDV* chains, such as *TRDV1* and *TRDV3*, decreased upon maturation towards effector fates (Fig. 4A). In contrast, no clear differences were observed in the usage of these *TRDV* gene segments between type 1, type 3, and type 2 effector cluster (Fig. 4A, right panel). Importantly, the enrichment of *TRDV2* usage upon maturation was accompanied by a decrease of N additions resulting in shorter CDR3 lengths and higher level of publicity (Fig. 4B–D). Also here no preference for a certain effector fate could be observed (Fig. 4B–D, right panels). Similar results were obtained for CDR3γ (Supplementary Fig. 6A), with the exception of the CDR3 length of *TRGV9*-containing CDR3 sequences as this feature increased rather than decreased in the effector cells; this was due to an increased usage of *TRGJP* that is known to be significant longer than all the other *TRGJ* sequences[2,14,27] (Supplementary Fig. 6B). Of note, the number of N additions and the associated CDR3 length and level of publicity correlated, together with TdT expression, with the gestation time (Supplementary Fig. 7).

Thus overall, upon maturation towards effector fate, γδ thymocytes undergo clear enrichments in public CDR3 features (*TRDV* usage, N additions, CDR3 length, publicity) at comparable levels in the three effector clusters.

## Fetal Vγ9Vδ2 and nonVγ9Vδ2 γδ thymocytes show similar effector programming

Vγ9Vδ2 T cells are recently described as the main innate-like γδ T cell subset in human[2,15,22,23], while nonVγ9Vδ2 γδ T cells adapt a more adaptive-like role[2,20,22,23]. Here, we compared Vγ9Vδ2 T cells and nonVγ9Vδ2 γδ T cells from their transition of the immature/maturing stage towards the different effector fates. The fetal thymocytes were in general enriched for *TRGV8* and *TRGV9* usage both in immature/maturing and effector clusters (Fig. 5A). The increase in *TRDV2* (Fig. 4A) and *TRGV9* (Fig. 5A) usage upon maturation corresponded to an increase in Vγ9Vδ2 T cells, i.e., cells with a paired *TRGV9/TRDV2* TCR, both at RNA (Fig. 5B) and protein (Fig. 5C) level. Note that the percentage of Vγ9Vδ2 T cells detected by sc γδ TCR followed the same trend according to gestation time compared to the percentages obtained by flow cytometry (Fig. 1C).

Unlike the adaptive-like nature in adult nonVγ9Vδ2, but in line with our previous bulk fetal thymus (FT) data[28], both Vγ9Vδ2 and nonVγ9Vδ2 γδ T cells were prevalent in the three effector fates (at RNA and protein level) (Fig. 5D), with a slightly lower Vγ9Vδ2 frequency in the type 2 effector cluster (at RNA level) (Fig. 5D, left panel). Bulk RNA sequencing of sorted Vγ9Vδ2 and nonVγ9Vδ2 γδ thymocytes confirmed these findings: while both Vγ9Vδ2 T cells and nonVγ9Vδ2 T cells showed clear differences compared to αβ T cells (Supplementary Fig. 8A), no gene related to effector functions could be found to be differentially expressed between Vγ9Vδ2 and nonVγ9Vδ2 fetal γδ thymocytes (Fig. 5E). Thus in the fetal thymus, both Vγ9Vδ2 and nonVγ9Vδ2 γδ T cells are pre-programmed towards defined effector fates.

Note that among the enriched genes in αβ thymocytes (compared to γδ thymocytes) one could find genes such as *FOXP3, CTLA4, TIGIT,* or *SATB1* (Supplementary Fig. 8A) that are enriched in Treg cells[78–80], which is in line with the presence of a regulatory phenotype in this T cell compartment in the human fetus[81]. Analysis of public CDR3 features (N additions, CDR3 length and level publicity) among *TRDV2*-containing CDR3 sequences uncovered also striking similarities at the level of these TCR characteristics between effector Vγ9Vδ2 and effector nonVγ9Vδ2 T cells (Fig. 5F, Supplementary Fig. 8B). This analysis (Fig. 5F) also excluded that differences at the level of *TRDV2*-containing CDR3 sequences upon maturation (Fig. 4B–D) were due to the parallel increase of Vγ9Vδ2 T cells (Fig. 5B). Thus overall, despite the increase of the Vγ9Vδ2 TCR upon maturation towards the effector clusters, no clear differences could be identified between Vγ9Vδ2 and nonVγ9Vδ2 γδ thymocytes among the effector clusters, both at the level of functional profile and general TCR features.

## Type 1, type 3, and type 2 γδ thymocytes express different CDR3 sequences

While no differences in general CDR3 features could be identified between the different effector γδ T cell clusters in the fetal thymus (Fig. 4), we went further to investigate these effector clusters at the level of the CDR3 sequence itself. Analysis of different fetal public *TRDV2*-containing sequences[2,14,27,28] displayed an association with effector clusters (Fig. 6). The CACDTGGY(S)WDTRQMFF sequence was most enriched in the type 1 effector cluster while it was low in the type 2 cluster (Fig. 6A). In contrast, the CACD(Y)WGSSWDTRQMFF sequence showed low frequencies in the type 1 cluster but was highly enriched in the type 2 cluster (Fig. 6B). The type 3 cluster showed intermediate frequencies of these two public *TRDV2*-containing sequences (Fig. 6A, B). The frequency of a hydrophobic residue at position 5 of the *TRDV2*-containing CDR3[82,83] and the frequencies of the public *TRGV8*-containing CATWDTTGWFKIF and *TRGV9*-containing CALWEVQELGKKIKVF sequences[2,10,14,16,27,84] increased upon maturation but did not differ in frequency according to effector type (Supplementary Fig. 9). Thus, we identified public CDR3δ sequences that are associated with the functional effector programming in the fetal thymus.

## The public CATWDTTGWFKIF and CALWEVQELGKKIKVF γ chains pair with different public δ chains

Using the paired information at the single-cell level we analyzed whether the public γ chains use the same or different public *TRDV2*-containing δ chains to form a TCR in the effector γδ thymocytes. We found striking preferences in the pairing of public γ with public δ CDR3 sequences: the *TRGV8*-containing CATWDTTGWFKIF preferred to pair with CACDTGGY(S)WDTRQMFF, while the *TRGV9*-containing CALWEVQELGKKIKVF with CACD(TV/I)LGDT*WDTTRQMFF* (*with TRDJ3*) or CACD(V/I)LGDT*DKLIF* (*with TRDJ1*) (Fig. 6C). Furthermore, while the δ chains CACD(**Y**)WGSSDTRQMFF and CACDWGSSDTRQMFF (Fig. 6B) only differ by one extra amino acid in the first sequence, they showed an opposite pairing pattern with the public γ chains (Fig. 6C). Note that the sequences that preferentially paired with the public CALWEVQELGKKIKVF (*TRGV9*) have a hydrophobic amino acid at the fifth position of their CDR3 (bold and underlined): CACD(T**V/I**) LGDT*WDTTRQMFF*, CACD(V/I)**L**GDT*DKLIF*, CACD(T/Y)**W**GSSDTRQ MFF. In contrast, the public CATWDTTGWFKIF (*TRGV8*) pairs with *TRDV2*-containing CDR3 that have a neutral glycine at their 5th position: CACDT**G**GY(S)WDTRQMFF and CACDW**G**SSDTRQMFF. Thus the two main fetal public γ chains show a strong preference for different public δ chains to form a γδ TCR.

## Lineage tracing distinguishes developmental pathways for effector clusters

The observation that distinct *TRDV2*-containing CDR3 sequences are enriched in different fetal γδ thymocyte effector clusters indicates

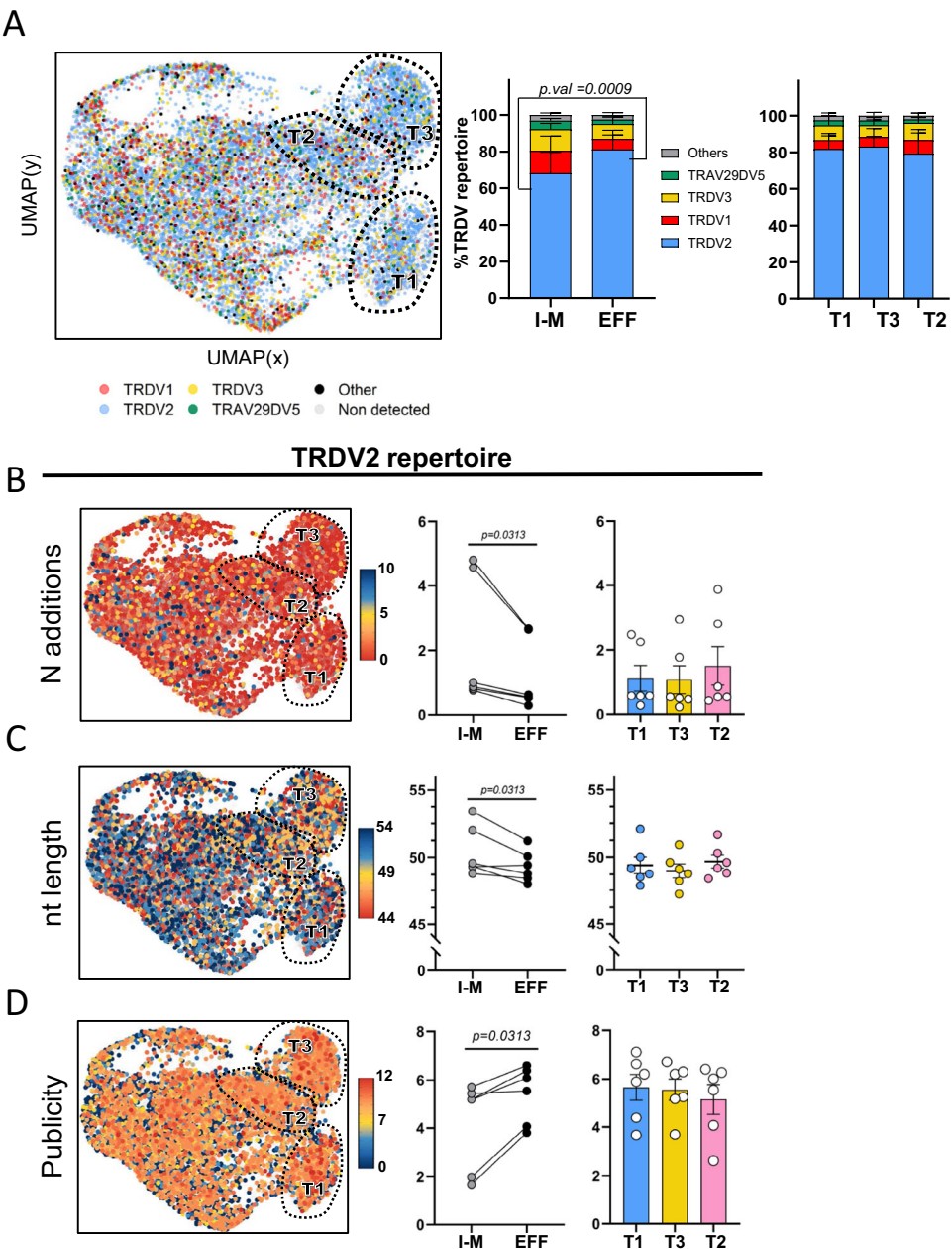

**Fig. 4 | Type 1, type 3, and type 2-like γδ thymocytes are enriched for public CDR3 features upon maturation.** γδTCR CDR3 regions were amplified from cDNA of fetal thymus sc RNA-seq libraries (*n* = 6). CDR3 data and transcriptomes were matched via cellular barcodes. **A** UMAP plot (left) and bar plots (center and right) displaying *TRDV* gene segment usage. Cells without TRD chain detected are displayed as light gray dots in the UMAP. "Others" groups: *TRDV4, TRDV6, TRDV7*, and *TRAV38-2DV8* variable chains. Percentages displayed in bar plots correspond to the mean value of the 6 thymic samples. **B** UMAP plot (left) and dot plots (center and right) display number of N additions (UMAP) and mean number of N additions (bar plots) in cells with *TRDV2*-containing CDR3 sequences. UMAP upper limit of the color scale represents sequences containing 10 or more N additions. **C** UMAP plot (left) and dot plots (center and right) display CDR3 nucleotide (nt) length (UMAP) and mean CDR3 nt length (dot plots) in cells with *TRDV2*-containing CDR3 sequences. UMAP lower/upper limits of the color scale represents sequences

containing 54/44 or more/less nucleotides, and they were established based on 0.8/0.2 quantile calculated on all the *TRDV2* CDR3 chains. **D** UMAP plot (left) and dot plots (center and right) display CDR3 publicity levels (UMAP) and mean CDR3 publicity values (dot plots) in cells with *TRDV2*-containing CDR3 sequences. Cells with a publicity level of 0 (navy blue color in UMAP) have a private *TRDV2* sequence which is only present in one sc TCR-seq library. "Immature/Maturing" group: c1,c2,c5,c7,c8,c9,c10,c11 in Fig. 2A. "Effector" group: c3,c4,c6 clusters in Fig. 1A. "Type 1", "Type 3", and "Type 2-like": c4, c3, and c6, respectively, in Fig. 1A (**A**–**D**). Bar plots/Dot plots are means ± SEM. Data in **A** was analyzed with ordinary two-ANOVA followed by Sidak's multiple comparison test, in **B**, **C**, **D** I-M and EFF values were compared by Wilcoxon-matched pairs signed-rank test, while T1, T2, and T3 groups were compared by Ordinary one-way ANOVA with Tukey's multiple comparisons test as Post Hoc test. Source data are provided as a Source data file. See also Supplementary Figs. 6 and 7.

that different TCR signaling strengths guide immature γδ thymocytes towards different effector fates. To explore this further, we studied the potential developmental pathways by performing trajectory analysis using the *Slingshot* R package. This new analysis, focused on TRDV2 cells (at CDR3 level), identified three distinct

lineage pathways ending in the three effector clusters, with an early split between the type 1 (T1 lineage, L1) versus the type 2 (T2 lineage, L2) and type 3 (T3 lineage, L3) pseudotimes (Fig. 7A and Supplementary Fig. 10). Assessment of the expression pattern of genes over the T1 lineage identified early onwards markers (i) induced after TCR

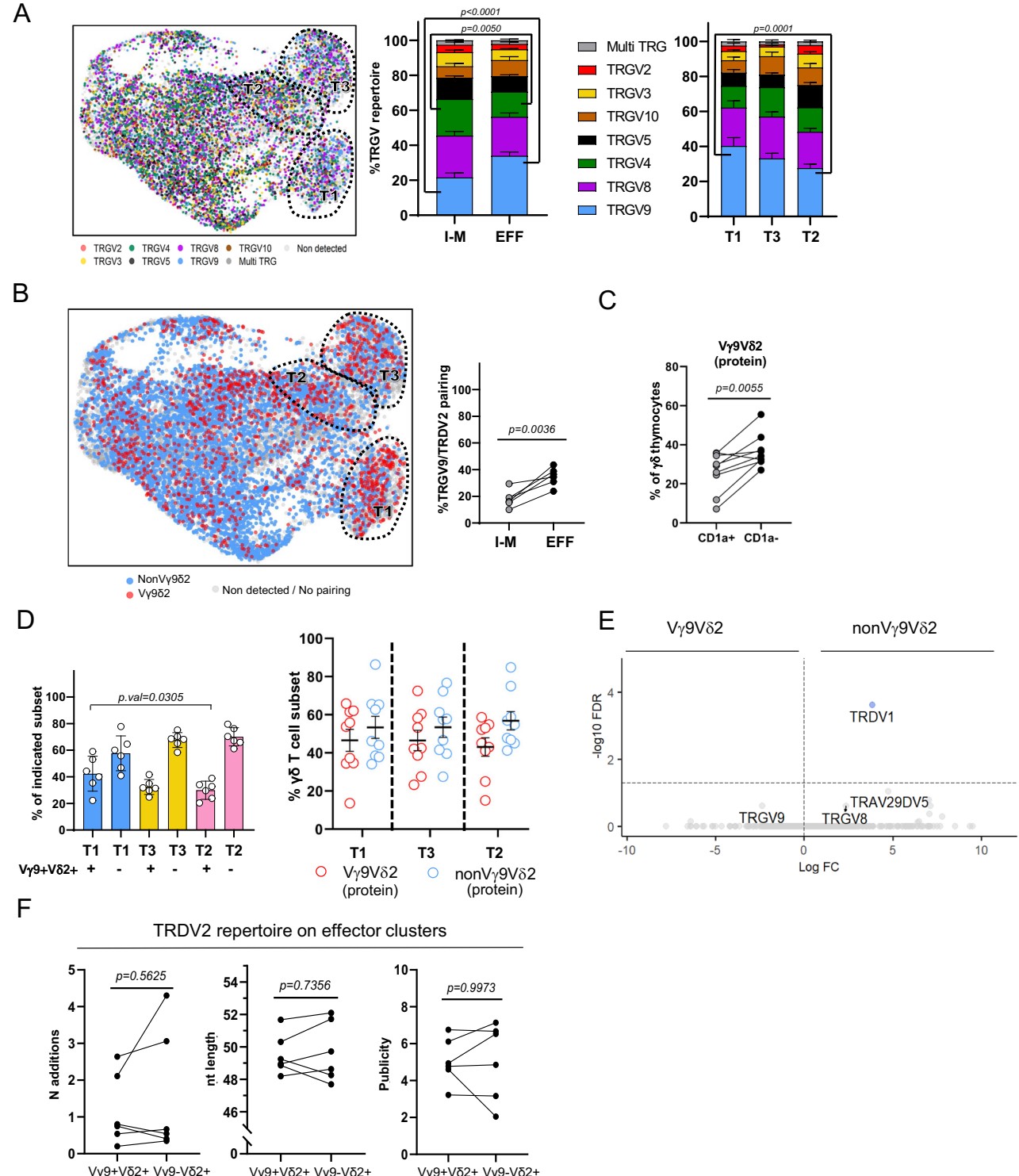

**Fig. 5 | Fetal Vγ9Vδ2 and nonVγ9Vδ2 γδ thymocytes show similar effector programming.** γδ TCR CDR3 region was amplified from cDNA of fetal thymic sc RNA-seq libraries (*n* = 6). CDR3 data and transcriptomes were matched via cellular barcodes. **A** UMAP plot (left) and bar plots (center and right) displaying *TRGV* gene segment usage. Cells without TRG-detected chain are displayed as light gray dots in the UMAP. **B** UMAP plot (left) and dot plot (right) displaying Vγ9Vδ2 cells (left) and percentage of Vγ9Vδ2 cells (center and right); Vγ9Vδ2 T cells are identified based on the sc TCR (CDR3) data. **C** Paired dot plot indicating the % of Vγ9Vδ2 thymocytes in the immature (CD1a⁺) and mature (CD1a⁻) subsets evaluated at protein level from 9 different subjects. **D** Dot plots indicating the % of Vγ9Vδ2 and nonVγ9Vδ2 thymocytes in the distinct effector clusters, at transcriptomic level (left panel) and at protein level (right panel). **E** Volcano plot illustrating differentially expressed genes

between Vγ9Vδ2 and nonVγ9Vδ2 thymocytes. Cell populations were sorted from different subjects than the ones used in the sc experiment (*n* = 3, age = 17 wk, 17 wk, and 19 wk of gestation). **F** Paired dot plots displaying number of N additions, nucleotide length and publicity levels of *TRDV2*-containing CDR3 sequences from Vγ9Vδ2 and nonVγ9Vδ2 thymocytes. Data in **A** was analyzed with ordinary two-ANOVA followed by Sidak's multiple comparison test, in **B**, **C** groups were compared by two-tailed paired *t*-test, while in **D** groups were compared by Ordinary one-way ANOVA with Tukey's multiple comparisons test as Post Hoc test. Error bars in **D** correspond to the SEM of 6 (right panel) and 9 subjects (right panel). "I-M" group: immature/maturing. "EFF" group: effector. "EFF" group: effector. "T1", "T3", and "T2" groups: Type 1, Type 3, and Type 2-like. Source data are provided as a Source data file. See also Supplementary Fig. 8.

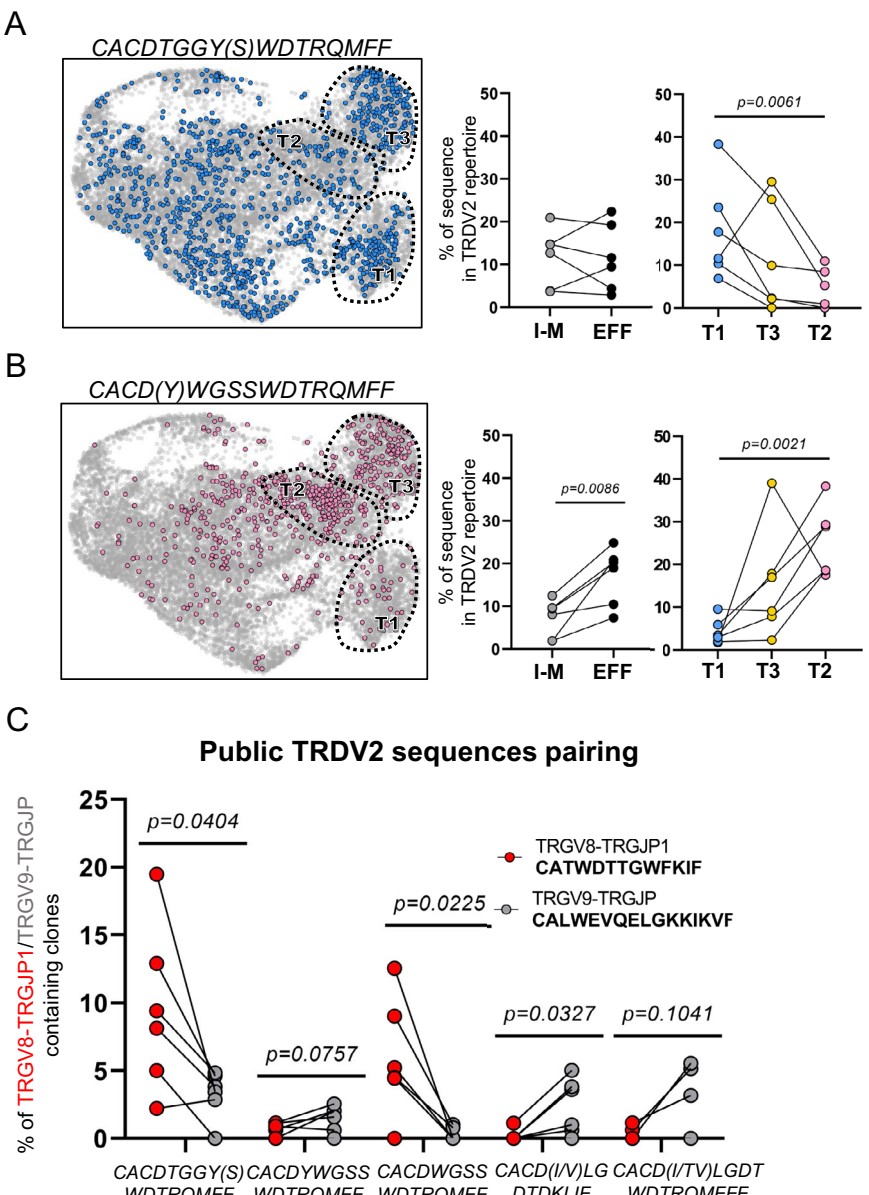

**Fig. 6 | Differential association of *TRDV2*-containing CDR3 sequences with effector modules and with public CDR3γ sequences in the human fetal thymus.** γδ CDR3 region was amplified from cDNA of fetal thymic sc RNA-seq libraries (*n* = 6). CDR3 data and transcriptomes were matched via cellular barcodes. **A** Plots indicate distribution (UMAP; left) and percentage (paired dot plots; center and right) of cells containing a CACDTGGYWDTRQMFF or CACDTGGYSWDTRQMFF *TRDV2* sequence. **B** Plots indicate distribution (UMAP; left) and percentage (paired dot plots; center and right) of cells containing a CACDWGSSWDTRQMFF or CAC-DYWGSSWDTRQMFF *TRDV2* sequence. Light gray dots in the UMAPS indicate cells with CDR3δ sequences different than the highlighted ones or cells without detected CDR3δ sequence. **C** Cells containing different public *TRDV2* sequences were split in two groups based on their CDR3γ pairing: cells paired with the public

CALWEVQELGKKIKVF *TRGV9-TRGJP* chain (gray color) and cells paired with the public CATWDTTGWFKIF *TRGV8-TRGJP1* chain (red color). The percentages were calculated based the total number of cells containing a *TRDV2* chain paired with *TRGV9-TRGJP* or *TRGV8-TRGJP1* gamma chains, respectively. I-M vs EFF comparison in **A**, **B** was analyzed with two-tailed paired t-test, while comparisons between effector clusters were performed by Ordinary one-way ANOVA for matched data with Holm-Sidak's multiple comparisons test as Post Hoc test when data followed normality or Friedman test followed by Dunn's test as Post Hoc test when data did not follow normality. Data in **C** was analyzed by two-tailed paired *t*-test. Error bars in **A**, **B**, **C** correspond to the SEM. "I-M" group: immature/maturing. "EFF" group: effector. "T1", "T3", and "T2" groups: Type 1, Type 3, and Type 2-like. Source data are provided as a Source data file. See also Supplementary Fig. 9.

activation (*PDCD1*, *MME* (CD10), *TOX2, ZNF683* (Hobit))[85,86], (ii) involved in T cell co-stimulation (*CD27* (TNFRSF7), *CD28, TNFRSF4* (OX40L), *TNFRSF9* (4-1BB), *TNFRSF18* (GITR))[53,87–89], and (iii) in TCR signaling (*LAT2, PTPN6*)[90–93] (Fig. 7B). These observations are consistent with an important role for strong TCR signaling in mouse type 1 γδ development[53,57,94–96]. Of note, *LEF1*[97] a gene described to promote mouse type 1 γδ T cells, was enriched as well in T1 lineage (Fig. 7B). The expression of these genes in the T1 pathway was followed by the strong induction of PLZF (*ZBTB16*)[98] (Supplementary

Fig. 10) which was downregulated again towards the final effector fate (Fig. 3D). Such early strong signals of TCR activation could not be observed in type 2 and type 3 pseudotime pathways (Fig. 7B, middle and right). However, at the end of the all the three effector trajectories, similar levels of *CD69* and *NR4A1* could be observed, suggesting some common degree of TCR activation[99,100] at these final phases (Supplementary Fig. 10). In addition to that, at this end phase, the type 2 trajectory showed strong expression of *CD5* transcripts, known to be associated with TCR signaling[86,87,90], which can explain

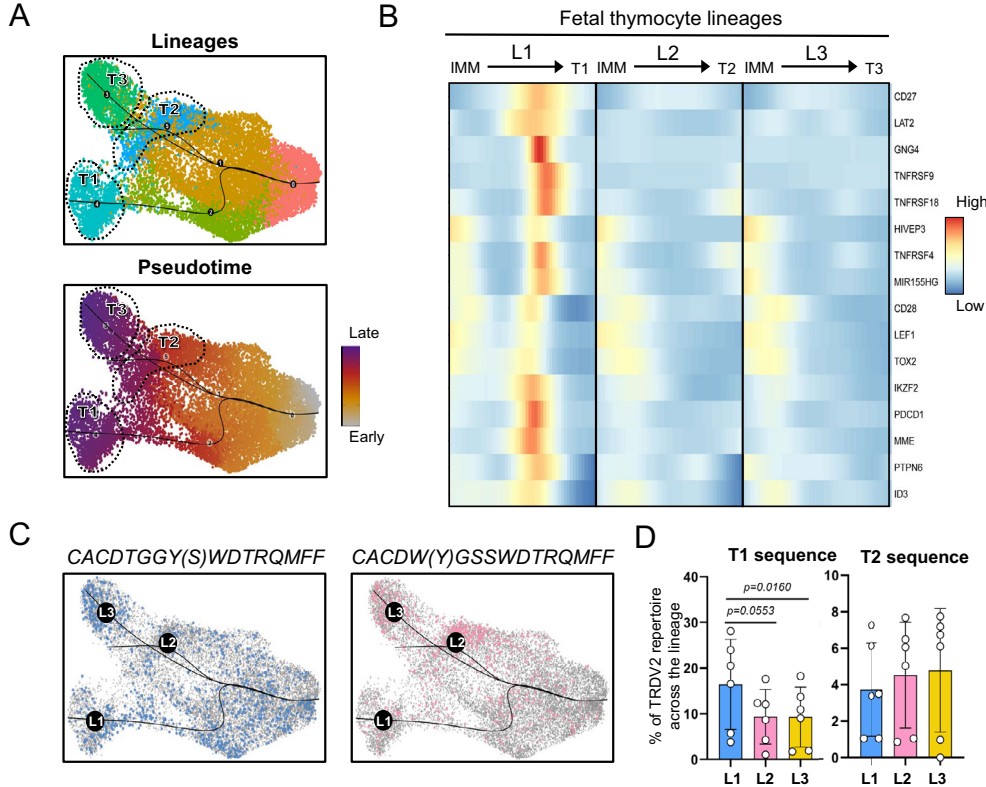

**Fig. 7 | Lineage tracing distinguishes developmental pathways for effector clusters. A**, top panel: UMAP plot of TRDV2 γδ thymocytes with Slingshot trajectories. This focused analysis identified 6 clusters with clusters 4,3 and 5 corresponding to Type 1, 3 and 2-like effector profiles respectively. **A**, bottom panel: UMAP plot of TRDV2 γδ thymocytes displaying pseudotime values across lineages. **B** Heatmap of smoothed scaled gene expression of selected genes differentially expressed across Type 1 lineage pathway (L1) compared to the other lineages (L2 and L3); note that the endpoints (effector clusters) themselves were not included in this analysis. **C** UMAP plot of γδ thymocytes with Slingshot trajectories and cells colored according to the indicated public *TRDV2* chain they express. **D** Bar plots showing the abundance of Type 1 associated *TRDV2* sequence (CACDTGGY(S)WDTRQMFF) (Left) and Type 2 associated *TRDV2* sequence (CACD(Y)WGSSWDTRQMFF) (Right) across the distinct lineages (excluding the effector clusters) in the 6 fetal subjects. Data in **D** was analyzed by Ordinary one-way ANOVA for matched data with Holm-Sidak's multiple comparisons test as Post Hoc test. Error bars in **D** correspond to the SEM. Source data are provided as a Source data file. See also Supplementary Fig. 10.

the high levels of PLZF identified previously in the type 2 effector cluster (Supplementary Fig. 10; Fig. 3D). Finally, type 2 and type 3 lineages expressed in a more continuous way markers such as the naïve T cell marker *SELL* (CD62L), *GIMAP4*, *IL7R*, and *RUNX1*[48,101] (Supplementary Fig. 10). The downregulation of the expression of *RUNX1* in the type 1 pathway corresponds with the timing of the early and strong TCR signaling (Supplementary Fig. 10), which is in line with the described downregulation of this TF after TCR signaling in mouse CD4 αβ T cells[102], while the IL7R expression becomes highest in the type 3 effector fate cluster (Supplementary Fig. 10), in line with the importance of this receptor for the expansion of these cells[59]. Of note, the expression pattern of *RUNX1* and *ZBTB16* (PLZF), enrichment across the type T3 and T2 lineages, are in line with the presence of a Runx1-bound *ZBTB16* enhancer[103].

Next, we analyzed the presence of the type 1 (Fig. 6A) and type 2 (Fig. 6B) CDR3 sequences along the different lineages identified by the trajectory analysis (Fig. 7C). This analysis indicated that the type 1 effector-associated CDR3 sequence CACDTGGY(S)WDTRQMFF was already enriched in the immature/maturing lineage 1 γδ thymocytes (Fig. 7D).

Thus, trajectory analysis showed that type 1 and type 2/3 lineages split early onwards during the development of immature γδ thymocytes and that the T1 lineage thymocytes go through several early developmental stages that are consistent with strong TCR and associated co-stimulation signaling.

## The pediatric thymus generates a small Vγ9Vδ2 type 1/type 3 effector subset

In order to complement our findings regarding γδ thymocyte development in the human fetus, scRNA + TCR sequencing was applied on sorted post-natal γδ thymocytes from three children (4.0 y, 4.5 and 11 years old). 7295 cells were subjected to downstream analysis (average gene number = 1350; average Unique Molecular Identifier or UMI = 3695) (Supplementary Fig. 11A). Unsupervised graph-based clustering on UMAP representation identified 10 distinct clusters (Fig. 8A) with similar distribution across subjects (Supplementary Fig. 11D). Immature populations were identified in c5, c8 (Fig. 8A; Supplementary Fig. 11B and C). DGE analysis (Supplementary Fig. 11C; Supplementary Data 4) identified common cell states with the fetal thymus, such as cycling cells (c8), cells with a type I interferon response gene signature (8), and cells with a TCR activation/co-stimulation profile (c4) (Supplementary Fig. 11C; Supplementary Data 4). A series of markers of which we showed previously differential expression by bulk RNA sequencing between fetal and post-natal thymus[28] were confirmed to be differentially expressed in the UMAP plots (Supplementary Fig. 14). Interestingly, c6 and c10 contained features associated with an effector profile, as they were enriched for NK receptors and cytotoxic related molecules, thus linking it to type 1 immunity (Fig. 8B and Supplementary Fig. 11C). Furthermore, these two clusters were the only ones presenting a more uniform egress potential (Fig. 8C, left panel). Cluster 6 cells expressed mainly *TRDV1*-containing CDR3 sequences,

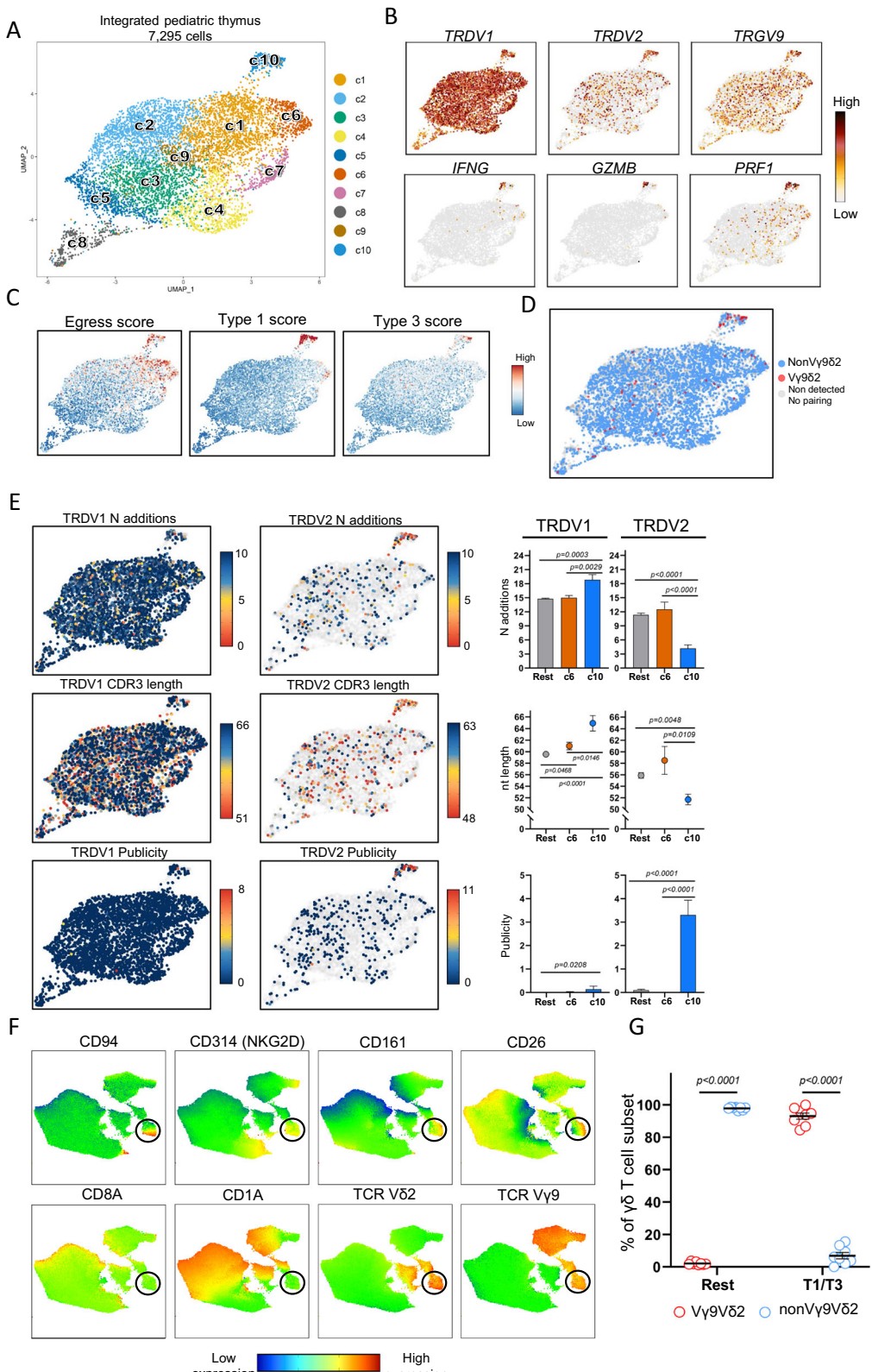

like most of the other clusters in the post-natal thymus (Fig. 8B, E), were enriched for *NCR3* (NKp30; validated at protein level, supplementary Fig. 12A and D), *ZNF683* (Hobit) and *FCER1G* and expressed, though at lower levels than cells from c10, the chemokine *CCL5* and the cytotoxic mediator granulysin (*GNLY*) (Supplementary Fig. 11C; Supplementary Fig. 14). Since all these features are associated with the anti-cancer Vδ1+ NKp30+ γδ T cells in the blood of adult subjects[104–108],

we propose that c6 in the post-natal thymus can be the source of these cells in the adult circulation.

Cluster 10 highly expressed type 1-related effector molecules such as granzymes (*GZMA, GZMB, GZMK, GZMM*), *PRF1* (perforin) and *IFNG* (Fig. 8B, C, Supplementary Fig. 11C; Supplementary Fig. 14), with a minor subset of cells containing transcripts associated with type 3 immunity although no *IL17A* or *IL17F* expression was detected (Fig. 8C;

**Fig. 8 | The pediatric thymus generates a small Vγ9Vδ2 type 1/type 3 effector subset.** Single γδ thymocytes from pediatric thymus samples ($n = 3$; 4.0, 4.5, and 11.0 years) were sorted in three independent experiments. After quality control 7295 cells were subjected to downstream analysis. **A** UMAP visualization of the annotated clusters obtained after integration of the samples. **B** Expression levels of selected genes in the UMAP plot. **C** Projection of egress (left), Type 1 (middle), and Type 3 (right) module scores computed using different set of marker genes (Supplementary Data 5). Each cell is colored based on their individual score. **D** UMAP plot displaying Vγ9Vδ2 cells based on CDR3 data. Cells without paired TRD and TRG chain are displayed as light gray dots in the UMAP. **E**, left and center: UMAP plots displaying N additions (top), CDR3 nt length (middle), and CDR3 publicity levels (bottom) of *TRDV1*-containing (left) and *TRDV2*-containing (center) sequences. Light gray dots in the UMAP plots indicate cells with CDR3δ segments using different *TRDV* than *TRDV1* (left) or *TRDV2* (center) gene segments. Upper limit color scale of N additions UMAP plots represent sequences

containing 10 or more N additions. Cells with a publicity level of 0 (navy blue color in UMAP) have a private *TRDV2* sequence which is only present in one sc TCR-seq library. **F** UMAP plots derived from flow cytometry data indicating the expression levels of distinct surface markers. UMAP plots include 28,421 live CD3+ γδTCR+ thymocytes from 5 different pediatric thymus samples. Black circles are used to facilitate the location of the small effector blended Vγ9Vδ2 cluster. **G** Dot plot indicating the frequency (%) of Vγ9Vδ2 and nonVγ9Vδ2 thymocytes at protein level in the T1/T3 (effector blended) cells (CD1a⁻CD94⁺CD161hi + CD26⁺) and the rest group (CD1a⁻CD94⁻CD161⁻). **E, G** Displayed error bars are means ± SEM. Data in **E** was analyzed by Kruskal–Wallis test followed by two-tailed Dunn's multiple comparisons test. In **G**, comparisons between Vγ9Vδ2 and nonVγ9Vδ2 subsets in the distinct effector clusters were performed using two-tailed Wilcoxon-matched paired tests. "Rest" group: all clusters except c6 and c10 in **A**. Source data are provided as a Source data file. See also Supplementary Figs. 11–13.

Supplementary Fig. 14). Strikingly, c10 was the only cluster that was enriched for the expression of *TRDV2* and *TRGV9* genes (Fig. 8B) and, as shown by sc TCR sequencing, was highly enriched for Vγ9Vδ2 T cells (Fig. 8D and Supplementary Fig. 13A). In line with the described egressing potential of cells in c10, flow cytometry analysis confirmed a mature profile (CD1a-) of pediatric Vγ9Vδ2 thymocytes (Supplementary Fig. 12C). Therefore, in contrast to fetal Vγ9V2 thymocytes (Fig. 5D, E), post-natal Vγ9Vδ2 thymocytes appear to express a distinct effector profile compared to nonVγ9Vδ2 γδ thymocytes. This finding was validated by bulk RNA sequencing on sorted Vγ9Vδ2 and nonVγ9Vδ2 thymocytes from three different thymuses (ages: 4 months, 1 year, 8 years) confirming the expression of type 1/3 immunity-associated genes in the Vγ9Vδ2 subset (Supplementary Fig. 13B). Of note, c10 cells exhibiting a potential dual-effector profile (type 1 and type 3) were highly enriched for *TRDV2*-containing CDR3 sequences with relative low N additions and high levels of publicity (Fig. 8E). No 'fetal-like' features (low N additions, high publicity) could be found in any of the *TRDV1*-containing CDR3 sequences of the post-natal thymus, including sequences of effector c6 (Fig. 8E). Note that Vγ9Vδ2 T cells, with a few of them containing low N additions/high publicity, were dissipated among the immature *TRDV1*-enriched clusters (Fig. 8B, Fig. 8D–E). Cluster 10 also contained *TRGV9* sequences with a lower number of N additions and higher levels of publicity (Supplementary Fig. 13C). Finally, we validated at the protein level the prevalence of a blended type 1/3 Vγ9Vδ2 cluster in the pediatric thymus (Fig. 8F–G; Supplementary Fig. 12A). Interestingly, in contrast to the *TRDV1*-enriched c6, c10 was not enriched for the NKR NKp30 (*NCR3*) but rather for the NKR CD94/NKG2A, both at scRNA (Supplementary Fig. 14) and protein (Fig. 8F) level, indicating that the expression of this inhibitory NKR on adult blood anti-cancer Vγ9Vδ2 T cells[109] is already programmed in the thymus (Fig. 8F; Supplementary Fig. S11C; Supplementary Fig. S12A, B; Supplementary Fig. 14).

Taken together, the post-natal thymus shows a much more limited generation of programmed effector γδ thymocytes compared to the fetal thymus, with a small but highly specialized Vγ9Vδ2 effector cluster containing features of both type 1 and type 3 immunity.

## Discussion
The difference between mouse and human γδ T cells together with the limited access to human thymic material have hampered the understanding of human γδ T cell development. Here, we applied scRNA+TCR sequencing on sorted γδ thymocytes from fetal and pediatric thymuses in order to obtain insight into the intra-thymic development of the 'third way of protection'[1] in human.

The mature effector γδ thymocytes of the human fetal thymus were divided into three discrete effector fates based on their gene expression profiles: type 1, type 3, and type 2-like. The type 1 immunity effector program included genes such as the signature cytokine *IFNG*, the transcription factor *TBX21* (Tbet), and cytotoxic mediators such as

*PRF1* (perforin) and granzymes (*GZMA, GZMK*); type 3 immunity included a series of genes previously described to be associated with mouse type 3 γδ T cells[51,54–60] and included the expression of *IL17A* and *IL17F*; and finally, the last effector fate we rather termed type 2-'like' as, while enriched for a series of genes associated with this type of immune cells (e.g., *ICOS, IL4R, CCR4*), it lacked the programmed expression of a signature cytokine such as IL4. In the mouse, a γδ T cell subset, known as iNKT-like γδ cells or γδNKT cells, has been described that readily produces IL4 and that are enriched for the expression of the Vγ1 and Vδ6 chains[2,71,110–113]. While these features (high IL4 expression, enrichment of particular Vγ/Vδ chains) were not observed in the human fetal thymic type 2 effector profile, other features are shared: γδNKT cells have also been described to have a fetal thymic origin[111], a high PLZF expression (which has been shown to be important for their development[110]), and a high level of CD4 expression[71,114]. At the level of NKR expression, we found different expression patterns among the three effector fates: CD161 (*KLRB1*), an inhibitory NKR which is directly involved in the control of the effector γδ T cells in the fetal periphery[49], and NKp30 (*NCR3*) showed the highest expression on the type 3 effector subset, while type 1 effectors were specifically enriched for NKG2A/CD94 (*KLRC1/KLRD1*) and NKG2D. These NKR expression patterns suggest that the activation of the distinct γδ effector programs in the periphery is modulated by different NKR ligands[108]. For example, the high expression of the inhibitory NKG2A/CD94 NKR on type 1 cells may regulate the strong TCR signaling in this effector module. Human fetal type 3 γδ immunity, most abundant at the earliest gestation times investigated (14 wk), may play important physiological roles[115–117] and type 1 γδ immunity, showing increased proportion among γδ effectors during gestation (14–22 wk), may serve protection against viruses such as congenital cytomegalovirus infection that are showing increased placental transfer during this same gestation period[10,118]. Finally, type 2-like γδ cells can be important in providing help in the production of natural antibodies[62,119] by the fetus, which can be especially relevant in the period when there is no placental transfer yet of maternal antibodies[120].

Vγ9Vδ2 T cells are regarded as the main innate-like γδ T cells in human, while nonVγ9Vδ2 γδ T cells display adaptive features[2,22–24,121]. Therefore, it was surprising to observe that fetal effector Vγ9Vδ2 and nonVγ9Vδ2 γδ thymocytes did not show clear differences, both at the level of functional profile and general TCR features. Thus, we did not observe an association of particular *TRGV* (Vγ) usage with a particular function in the human fetal thymus as is commonly proposed for the waves of mouse fetal γδ T cell subsets, such as Vγ5+ γδ T cells producing IFNγ while Vγ6+ γδ T cells making IL17[58,122,123]. Compared to the human fetal thymus, the human pediatric thymus showed a completely different cell atlas. Here, Vγ9Vδ2 T cells were highly grouped in a small effector cluster that contained programmed features of both type 1 and type 3 immunity; type 2-like thymocytes were not identified in the post-natal thymus. These 'type 1/3' post-natal Vγ9Vδ2

thymocytes may contribute to the small fraction of IL17+IFNγ+ adult blood Vγ9Vδ2 cells that could be observed upon TCR+ IL23R stimulation in vitro[27,124] and may explain the switch of an 'IFNγ phenotype' towards an 'IL17 phenotype' in blood Vγ9Vδ2 T cells under pathological conditions[125–128]. Recently, Tan et al. identified in cord and adult blood by scRNA+TCR sequencing a type 3-associated cluster (though *IL17* transcripts could not be detected) to be associated with γδ T cells expressing semi-invariant Vγ9Vδ2 TCRs[129], and, together with scRNA data from embryonic thymic organogenesis[29,130], suggest that this subset of Vγ9Vδ2 T cells find their origin in the <10 wk embryonic thymus and persist till adulthood[129]. Here, we found that both Vγ9Vδ2 and nonVγ9Vδ2 fetal γδ thymocytes can develop towards a type 3 effector fate as well as toward type 1 or type 2-like effector fates. It is possible that only the fetal thymus-derived type 3 Vγ9Vδ2 T cells persist in the blood circulation, while type 3 nonVγ9Vδ2 and fetal type 1 and type 2 (either Vγ9Vδ2 or nonVγ9Vδ2) either seed the distinct peripheral tissues or disappear from the periphery upon aging. It was concluded by Tan et al. that type 3 cluster γδ thymocytes cannot be found anymore in >10 wk embryos[129,130], while we show the clear identification (including the expression of *IL17A* and *IL17F*) of this cluster at 14–17 wk of gestation. However, the conclusion by Tan et al. was based on scRNA data from total thymocytes without isolation of γδ thymocytes[130] or another way to unambiguously identify γδ T cells[29,30]. It is likely that the increase of αβ thymocytes from 10wk onwards in the fetal thymus[29] compromised the identification of the type 3 γδ thymocyte cluster based on the expression of TRD and/or TRG genes[129]. Thus overall, human γδ thymocytes follow distinct developmental pathways evolving with age which probably reflect varying needs of the developing body and environmental encounters. The pediatric thymus harbors a small Vγ9Vδ2 TCR-enriched population with a mixed type 1/type 3 effector potential that, upon egress from the thymus, can respond rapidly to an infection. In contrast, the fetal thymus provides three ample clusters of specialized effector γδ thymocytes (type 1, type 3, type 2-like) which would cover a wider range of prompt needs for the developing fetus which might be linked to infections but also to physiological functions during growth in utero.

The variation present in the human fetal γδ TCR repertoire[2,28] allowed us to obtain insight into the development of the three effector fates. Surprisingly, all three fetal thymic effector types decreased upon maturation their number of N additions, decreased their CDR3 length and increased the level of publicity of their TCRs. Interestingly, the Vγ9Vδ2 T cells of the small type 1/type 3 effector cluster in the postnatal thymus appeared to undergo similar enrichments upon maturation, in contrast to the Vδ1+NKp30+ post-natal cluster. During the transition from DP (double positive) towards SP (single positive) stage in αβ thymocyte maturation, the CDR3 of the α and β chain becomes shorter which is related to MHC-imposed structural constraints[131,132]. The reason for the preference for short public TCRs upon MHC-independent γδ thymocyte maturation towards effector fates is unclear. Since these sequences are encoded by germline-encoded gene segments or only contain a very low number of N additions, the position of the amino acids within their CDR3 is less variable and may provide the 'right' TCR signal. A possible example is the presence of a hydrophobic amino acid at position 5 of phosphoantigen-responsive *TRDV2*-containing CDR3 sequences[82,83], of which the frequency increased upon maturation towards effector fates in the fetal thymus and which showed preferential pairing with the public *TRGV9*-containing CALWEVQELGKKIKVF CDR3. When more N additions are present, this can lead to, besides an increase in CDR3 length as such, a displacement of the germline-encoded CDR3 residues thus decreasing the chance to have a germline-encoded hydrophobic residue at position 5 of the CDR3δ sequence. Finally, shorter CDR3 length may influence the position of regions outside the CDR3 (such as the hypervariable region 4, HV4) of the TCR and thus the interaction with butyrophilins in the human thymus[5,99,133]. The relative contribution to the TCR signal in this setting of the CDR3 versus non-CDR3 TCR regions remains to be determined, but the significant changes observed here at the level of the CDR3 upon thymic maturation highlight the importance of the CDR3 in the maturation towards effector fates in the human fetal thymus.

The identification of different CDR3 sequence enrichments in the three fetal thymic effector clusters combined with their pseudotime developmental trajectories strongly suggest the presence of three developmental pathways. The fact that the three effector clusters did not show differences in CDR3 N additions, argues against a different timing in the generation of particular CDR3 sequences (as observed in the mouse model) as a possible explanation for their association with particular effector fates. We rather propose that the CDR3 sequence contributes to a difference in TCR signal strength during maturation and thus, together with signaling from other receptor types such as NKR and/or cytokine receptors and/or precursor frequencies[59,74,75], to the type of effector fate. Type 1 and type 2/3 clusters split early onwards during the development of immature γδ thymocytes. Type 1 γδ thymocytes went then through several early stages of maturation associated with strong TCR and associated co-stimulation signaling, which is in line with the need for such signals for the development of type 1 γδ thymocytes in (genetically-modified) mouse models[53,95,96,134]. Furthermore, we identified genes (*TNFRSF9, XCL1, TNFRSF13*) across the human type 1 developmental pathway that are also highly expressed during the thymic Skint1-mediated and extra-thymic Btnl1-mediated TCR-dependent selection of mouse type 1 γδ T cells[89,135]. Of note, *TNFRSF9* (4-1BB) has been shown to be induced on human Vγ9Vδ2 T cells by phosphoantigens[136], consistent with a TCR-dependent and butyrophillin-dependent (BTN3A1, BTN2A1, BTN3A2)[5,133] regulation of this co-stimulatory receptor. Thus, despite the known differences between mouse and human γδ T cells[2,123,137] and the large difference in the timing of the development of the fetal/neonatal immune system[138,139], our observations during human fetal type 1 γδ thymic development are strikingly similar to what has been shown in mouse models. The type 3 and type 2 developmental pathways were largely shared but split at the final maturation stage. Overall, based on the expression patterns of TCR-signaling related markers[57,96,134,140] and of the transcription factor PLZF[103,110,134,141,142], we propose that differences in timing and strength in TCR signaling result in associated differences of the transcription factor PLZF that then guides the final thymic γδ effector fate.

In summary, we have generated a cell atlas of human γδ thymocyte development across fetal and post-natal life, from the most immature stages until programmed effector fates. This combined database of gene expression and detailed TCR information at the single-cell level has provided insight into γδ T cell development in the human thymus and provides a resource for further study.

## Methods

### Human fetal and post-natal thymus

Human fetal thymus samples (*n* = 11) were obtained from 14 to 22 week estimated gestational age elective pregnancy terminations carried out for socio-psychological reasons with approval of the Singapore Singhealth Research Ethics Committee. Women gave written informed consent for the donation of fetal tissue to research nurses who were not directly involved in the research, or in the clinical treatments of women participating in the study. All the donors were informed about the purpose of the research and there was no compensation offered for donation. All fetuses were considered structurally normal on ultrasound examination prior to termination and by gross morphological examination following termination. Human pediatric thymus (9 donors aged between 1 and 11 years) samples were obtained from children that underwent cardiac surgery with approval of the Medical Ethical Commission of the Ghent University Hospital (Belgium). Samples from the previous sources were collected after all participants

(when applicable, mothers/parents) gave written informed consent in accordance with the Declaration of Helsinki. Cell suspensions from fetal thymus and post-natal thymus samples were obtained as previously described[28].

## Flow cytometry and sorting of γδ thymocytes

For flow cytometry (assessment of percentage of γδ and Vγ9Vδ2 thymocytes) and associated cell sorting (FACS) of the samples used to generate the single-cell libraries, cells were thawed in complete medium, washed twice, labeled with Zombie NIR dye (0.5:100; BioLegend), and then subsequently stained with antibodies directed against CD3 (dilution 1:100 for flow cytometry and 2.5:100 for sorting; clone UCHT1; BV510 for flow cytometry or PB for sorting; BD Biosciences), TCRγδ (dilution 1:100 for flow cytometry and 15/100 for sorting; clone 11F2; APC (Miltenyi Biotec) for flow cytometry and PE (BD Biosciences) for sorting), TCRVγ9 (dilution 0.25:100 for flow cytometry and 0.625:100 for sorting; clone IMMU360; PE-Cy5; Beckman Coulter) and TCRVδ2 (dilution 4:100 for flow cytometry and 10:100 for sorting, clone IMMU389; FITC; Beckman Coulter). For sc experiments, CD3$^+$γδTCR$^+$ thymocytes were sorted (mean purity 98% of living cells) on a FACS Aria III (BD Biosciences). For bulk RNAseq experiments, the CD3$^+$γδTCR$^+$ thymocytes were further sorted into CD3$^+$ γδTCR$^+$ Vγ9$^+$Vδ2$^+$ as "9Vδ2" (mean purity 95% of living cells), and CD3$^+$γδTCR$^+$ non(Vγ9Vδ2) as "non-Vγ9Vδ2" γδ T cells (mean purity 95% of living cells)[28]; αβ T cells (CD3$^+$TCRγδ$^-$) were sorted as well in parallel (all around 10,000 cells) on a FACS Aria III cell sorter (BD Biosciences), snap-frozen in liquid nitrogen, and stored at −80 °C for later RNA extraction. To validate the presence of the distinct populations identified in the single cell data the following antibodies were used: CD3 (dilution 1:100, clone UCHT1; BV510; BD Biosciences), TCRγδ (dilution 1:100, clone 11F2; APC; Miltenyi Biotec), TCRVγ9 (dilution 0.25:100, clone IMMU360; PE-Cy5; Beckman Coulter) and TCRVδ2 (dilution 4:100, clone IMMU389; FITC; Beckman Coulter), CD4 (dilution 1:100, clone SK3; BUV395; BD Biosciences), CD26 (dilution 2:100, clone M-A261; BUV496; BD Biosciences), NKG2D (dilution 3:100, clone 1D11; BV421; Biolegend), CD196 (dilution 2:100, clone 11A9; BV650; BD Biosciences), CD1a (dilution 2:100, clone HI149; BV711; Biolegend), CD278 (dilution 0.5:100, clone C398.4A; BV785; Biolegend), CCR4 (dilution 2:100, clone 1G1; PE; BD Biosciences), CD94 (dilution 2:100, clone DX22; PE-Cy7; Biolegend), CD161 (dilution 2:100, clone DX12; R718; BD Biosciences), CD8a (dilution 2:100, clone RPA-T8; APC-Cy7; BD Biosciences), NKp30 (dilution 2:100, clone P30-15; PE-Dazzle; Biolegend). In these protein validation experiments, measurements were taken from 9 distinct fetal samples and 8 infant samples that were thawed in complete medium and washed twice prior to staining. iFluor860 (infrared fixable viability dye) (dilution 0.05:100; AAT Bioquest) was used to gate on live cells. In all cases (FACS or flow cytometry experiments), the data were analyzed using FlowJo software under version 10 (Tree Star). To generate the UMAP plots in Fig. 3 and Fig. 8, we used the Flowjo plugin "UMAP" (v3.1) and in both cases we computed it by Euclidean distances with 2 components, a value of 15 for the nearest neighbors parameter and a value of 0.5 as minimum distance. For the UMAP of Fig. 2, dimensional reduction process involved the following cell surface markers: CD1a, CD94, CD161, CD4, ICOS, CCR4, CCR6, CD26, and NKG2D. For the UMAP of Fig. 8, we used the values from the following markers: CD1a, CD94, CD161, CD4, ICOS, CCR4, CCR6, CD26, NKG2D, TCRVδ2, and TCRVγ9. In this last case, we decided to include TCRVδ2 and TCRVγ9 markers to facilitate the visualization of the small Vγ9Vδ2 effector cluster.

## Single-cell RNA-seq and single-cell TCR (TRD/TRG)-seq libraries construction

Libraries for sc RNA and TCR sequencing were generated from 0.5–2×10$^4$ FACS-sorted γδ thymocytes from six fetal subjects and three children using the Chromium Single Cell 5′ Library Gel Bead and Construction kit as well as Chromium Single Cell V(D)J Enrichment Kit (10x Genomics, CA, USA) according to the user guidelines (v1 [PN-1000006] and v2 [PN-1000244] Chemistry, Single Cell V(D)J protocol number CG000086 and CG000331). Fetal sample selection included six fetuses with an estimated gestation time of 14 weeks, 15 weeks and 2 days, 16 weeks and 2 days, 17 weeks and 5 days, 21 weeks, and 22 weeks and 6 days, while post-natal thymuses were from patients with 4, 4 and a half, and 11 years of age, respectively. Measurements were taken from these distinct samples.

Single-cell TCR & gene expression libraries were generated according to Chromium Single Cell V(D)J protocol (10x Genomics). 2 µL of cDNA amplified and purified from GEMs ("Gel bead in EMulsion" droplets) were used to amplify γδTCR CDR3 sequences. Custom primers specific for *TRDC* and *TRGC* constant gene segments were designed for this purpose and were obtained from Eurogentec. In brief, for the first step in the enrichment of CDR3 sequences the custom primers TRGC: CAAGAAGACAAAGGTATGTTCCAG and TRDC: GTAGAATTCCTTCACCAGACAAG were used, while for the second target enrichment Cgamma 'inner': AATAGTGGGCTTGGGGGAAA-CATCTGCAT and Cdelta 'inner': ACGGATGGTTTGGTATGAGGCT-GACTTCT were used. The remaining cDNA was used for gene expression library construction according to 10X Genomics protocol instructions. Agilent Bioanalyzer High Sensitivity DNA chips were used to check quality control read-outs of sc RNA-seq and sc TCR-seq libraries using a Bioanalyzer 2100 machine (Agilent Technologies). Indexed libraries were pooled and sequenced on Illumina NovaSeq 6000 device from BRIGHTcore (Brussels Interuniversity Genomics High Throughput core) platform.

## Single-cell RNA-seq data processing

*CellRanger* (v3.0.2) software from 10x Genomics was used to demultiplex and map sequencing reads against the GRCh38 genome. Count matrices were loaded into R using ´read10x´ function from *Seurat* R package. All downstream analyses were implemented using R v4.0.3 and the package *Seurat* v3.2.3[143]. Low-quality reads were filtered using the cutoff nFeature_RNA > = 200, while the cutoff for maximal nFeature_RNA was manually set-up for each sample according to the samples cell distribution in order to exclude doublets. Percentages of mitochondrial genes were plotted as well, and outliers were removed to filter out dead cells. 'Cellcyclescoring´ function from *Seurat* package was used to assign cell cycle phase of cells in the datasets (G1, G2, or S). Integration vignette from *Seurat* v3.0 was followed to generate merged *Seurat* objects (FT: 6 fetal thymus samples & PNT: 3 pediatrical thymy) using the 'SCTransform' function[144] and regressing mitochondrial genes, cell cycle genes and *TRDV* & *TRGV* genes. Principal components (PCs) were calculated using 'RunPCA' and by using 'ElbowPlot' visualization, 20 dimensions were chosen as input for 'RunUMAP' function. UMAP representation was used to generate bidimensional coordinates for each cell. The k-nearest neighbors of each cell was computed using the ´FindNeighbors' function and this knn graph was used to construct the shared nearest neighbor (SNN) graph by calculating the neighborhood overlap (Jaccard index) between every cell and its k.param nearest neighbors. Finally, the ´FindClusters' function was used to cluster cells using the Louvain algorithm based on the same PCs as RunUMAP function (algorithm resolution FT = 0.3 & PNT = 0.5). Cluster algorithm resolution was chosen after analyzing the evolution of the clusters at different resolutions with *clustree* R package (v.0.4.3). The Differential gene expression analysis comparing gene expression of each cluster to all the others was performed by the ´FindAllMarkers' function using Wilcoxon-Rank sum test method. DEGs were selected based on an average log2-fold change (logFC) ≥ 0.2, a percentage of expression superior than 10% in at least one test cluster (min.pct ≥ 0.1), a difference higher than 15% in the fraction of detection between the two groups (min.diff.pct ≥ 0.15) and adjusted *p*-value inferior than 0.05 (based on Bonferroni correction using all genes in the dataset).

dittoSeq (v1.4.1) R package was used extensively to visualize Seurat object data.

## Module scores

Single-cell gene signature enrichment scores were calculated using the 'AddModuleScore' function with the default parameters in *Seurat*. Egress score was manually curated using previously described markers described to be involved in thymocyte egress to periphery[35,45,46,72]. Type 1 score was named "CTL" score in the original paper where it was defined[145] and type 3 score was termed "γδ17" score in the original paper [54].

## GO and pathway enrichment analyses

Gene ontology (GO) analysis was performed by *clusterProfiler* package (v4.0.5)[146]. The gene list was arranged by logFC (decrescent order) obtained after comparing effector fetal clusters with the rest of cells using 'FindMarkers' function from *Seurat* with a min.pct ≥10%. GSEA was run using gseGO function with default parameters and using Benjamini–Hochberg method to obtain p.adjusted values. Enrichment results were plotted using *ggplot2* R package (v3.3.5).

## Single-cell TCR-seq analysis

Sc TCR libraries were generated by using CellRanger vdj pipeline (v3.0.2). Integrated FT and PNT Seurat objects (gene expression data) were combined with their respective sc TCR-seq (TCR sequence) data based on shared 10× cell barcodes and following the script provided here: https://www.biostars.org/p/384640/. Only those cells expressing productive TCR sequences (γ and/or δ chain) were retained for data integration in the Seurat objects using the 'Addmetadata' function. *TRDV* and *TRGV* sequences were used to check N nucleotides (N additions) and publicity levels. Number of N nucleotides was obtained using junctional analysis website tool from IMGT® (international ImMunoGeneTics information system®) website (MP, 2003). Barcodes were kept as identifiers for the input of the website tool and later used to embed the junctional information again in the Seurat objects in the metadata file and they were subsequently plotted using ggplot2 package (v 3.3.2). Publicity of *TRDV* and *TRGV* sequences was established by comparing individually all the CDR3 sequences from each single cell datasets against the CDR3 sequences of the other single-cell datasets (9 subjects in total, 6 fetal and 3 post-natal). In order to strengthen the analysis of publicity levels of CDR3 sequences, we decided to increase the number of subjects in the different comparisons by including CDR3 repertoire data obtained previously by bulk TCR repertoire. This new bulk TCR data included previously published data[27,28] and also unpublished data, resulting in a series of γδ thymocyte repertoires of 10 different subjects (3 fetal thymus samples and 7 pediatric thymus samples). The CDR3 data of these 10 bulk TCR repertoires was originally divided in 2 files: data from sorted Vγ9Vδ2 thymocytes[14] and nonVγ9Vδ2 γδ thymoctyes[28]. Because the goal of the publicity analysis is to check whether a specific sequence is present in one subject, we decided to merge the two files (Vγ9Vδ2 and nonVγ9Vδ2) in single combined files. Using base and dplyr (v1.0.7) R packages the amino acid CDR3 sequences of each of the thymy from the sc Seurat objects (6 subjects in the fetal thymus dataset and 3 subjects in the post-natal dataset) were interrogated individually against the bulk TCR data. The results of this analysis ranged from publicity values of 0 (present only in the interrogated sc TCR data) to 19 (present in the sc TCR data of the 6 fetal thymus samples and 3 pediatric thymus samples and the 10 bulk TCR repertoires). Results were added back in Seurat objects as metadata and plotted using ggplot2 package.

## Lineage inference

Pseudotime trajectory analysis of fetal γδ thymocytes was performed with the *Slingshot* R package under version 2.0.0[147]. In order to remove confounding factors, we excluded cycling cells (G2 and S phase) and cells belonging to the type I IFN cluster following the same reasoning described previously in the literature[72]. Then, Principal components (PCs) were calculated using 'RunPCA´ and by using 'ElbowPlot' visualization, 20 dimensions were chosen as input for 'RunUMAP' which was performed for 5 dimensions (instead of the standard 2 dimensions to reduce the distortion generated by the process of dimensionality reduction that can influence the lineage tracing results). Lineages were computed after selecting the cluster with immature features (based on gene expression) as a root. The calculated trajectories were overlaid into the UMAP embeddings. Genes that varied across the Slingshot trajectories were investigated with tradeSeq R package under version 1.6.0[148], and were plotted as heatmaps of smoothed scaled gene expression using ´predictSmooth´ function from tradeSeq and pheatmap R package (v1.0.12). The code used to generate Slingshot object and the usage of tradSeq package was obtained from the following website https://nbisweden.github.io/workshop-scRNAseq/labs/trajectory/slingshot.html#Finding_differentially_expressed_genes.

## Bulk RNA sequencing

RNA derived from sorted cell populations (Vγ9Vδ2, nonVγ9Vδ2 γδ, αβ) was isolated using the RNAeasy micro kit (Qiagen, Cat. No./ID: 74004). RNA quality was checked using a Bioanalyzer 2100 (Agilent Technologies). Indexed cDNA libraries were obtained using the Ovation Solo RNA-Seq System (NuGen) following the manufacturer's recommendation. The multiplexed libraries were loaded on a NovaSeq 6000 (Illumina) using an S2 flow cell, and sequences were produced using a 200 Cycle Kit (Illumina, PN: 20028313). Paired-end reads were mapped against the human reference genome GRCh38 using STAR software (version 2.7.10a) to generate read alignments for each sample. Annotations Homo_sapiens.GRCh38.90.gtf were obtained from ftp.Ensembl.org. After transcript assembling, gene level counts were obtained using *HTSeq*d software. Differential expression was performed by using EdgeR quasi-likelihood running under the Degust platform. Only genes with a minimum count per million of 1 in each replicate were included. Volcano plots were generated using *EnhancedVolcano* R package (v1.10).

## Statistical analysis

All statistical analyses were performed using GraphPad Prism software (v8.0.2).

## Reporting summary

Further information on research design is available in the Nature Research Reporting Summary linked to this article.

# Data availability

The scRNA-seq seq data, scTCR-seq data bulk TCR seq data from sorted FT and PNT Vγ9Vδ2 and nonVγ9Vδ2 thymocytes, together with the bulk RNA seq data of sorted FT and PNT Vγ9Vδ2 cells, have been deposited in the GEO database under accession code GSE180059. Bulk RNA sequencing data of sorted FT and PNT nonVγ9Vδ2 γδ and αβ thymocytes were deposited previously[28] in the GEO database with accession number GSE128163. Source data are provided with this paper.

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

## Acknowledgements

We thank Anne Lefort (Brussels Interuniversity Genomics High Throughput core platform) for her outstanding technical support. This work was supported by the Fonds De La Recherche Scientifique (FNRS, J.0225.20) (D.V.), the Fondation Jaumotte-Demoulin (D.V.), an ARC (ULB) grant (D.V.), a Win2Wal grant from the Walloon region (Immupromat, 1910062) (D.V.) and the European Regional Development Fund (ERDF) of the Walloon Region (Wallonia-Biomed portfolio, 411132-957270) (S.G.). G. Sanchez Sanchez (Télévie), M. Papadopoulou (post-doctoral fellowship), and Y. Tafesse (FRIA) are supported by the FNRS. S.G. is a research director from the FNRS.

## Author contributions

G.S.S. designed and performed experiments, analyzed data, and wrote the manuscript; M.P. designed and performed experiments and edited the manuscript; A.A. and F.L. provided support for data analysis; Y.T., I.V., and S.P. performed experiments; A.M., J.K.Y.C., Y.F., and F.G. provided samples; B.V. provided samples and edited the manuscript; S.G. designed experiments and edited the manuscript; D.V. designed and supervised the research, analyzed data and wrote the manuscript.

## Competing interests

The authors declare no competing interests.
