## [Peer Review File · Nature Communications]

Single-cell analysis of human fetal and pediatric $\gamma\delta$ thymocytes reveals distinct functional thymic programmingREVIEWER COMMENTS

Reviewer #1 (Remarks to the Author):

In this manuscript, the authors present a comprehensive analysis of gamma-delta T cell subsets in human fetal and postnatal thymocytes using scRNAseq and TCR sequencing. While mouse gd T cell development has been well-studied, much less is known about gd T cell development in humans. Comparisons of gd T cells between mice and humans have been complicated by the lack of direct homology between T cell receptor chains. However, the underlying regulators of effector function appear to be mostly conserved, allowing more useful comparisons by transcriptomics studies.

Here, the authors have asked the question of whether TCR identity is linked to effector function during development in the human thymus. To answer this question, they defined multiple subsets (cluster) of gd T cells in fetal and pediatric human thymus by single cell transcriptome and TCR analysis. This approach built upon earlier studies by this group and others using bulk RNAseq and FACS, and provided a much more in-depth analysis of the molecular signatures carried by different cells regarding of their TCR chain identities. Surprisingly, different TCRg and TCRd chains were not correlated with effector function, as they are in mice. Instead, gd T cell subsets expressing any combination of TCRg and TCRd chains could adopt any of the three major functional programs. However, certain “public” CDR3 sequences were strongly correlated with effector function. This included the Vg9 Vd2 subset, which has been a subset of some contention. These results are consistent with an instructive role for TCR signal strength in functional programming during development, as occurs in mice, but with more flexible gdTCR chain requirements.

Importantly, the transcriptional signatures of gd T cell subsets in the fetal thymus suggest a developmental trajectory that is consistent with that seen in fetal mouse gd T cell development. The results of this study indicate an early strong TCR signal for Type I effector programming, followed by differential TCR signaling in Type II and Type III gd T cells in the more mature stages of development. These critical finding validate the relevance of studies of mouse gd T cell development and function, and eliminates concerns about the lack of conservation between mouse and human TCR chains in identifying effector function. It also clears the way for combining the power of mechanistic studies in mice with detailed molecular analysis in humans, in an iterative manner. Together, these results are a significant advance in the field, and should have a major impact on future studies of the development and function of gd T cells in both humans and mice.

Major points

1. A more comprehensive discussion of the recently reported paper in Science Immunology by Prinz and colleagues on human gd T cells is warranted given the similarities and differences of their findings in the thymus versus their findings in the periphery.
2. The identification of Type I and Type III gd T cell subsets are well supported. Indeed, the finding of IL-17A and IL-17F in the Type III subset in the fetal thymus supports a previous study that defined Type III gd T cells in cord blood. These cells which did not express IL-17 at the mRNA level (Prinz and colleagues), suggesting that in the periphery, stimulation is needed for these cells to elaborate their Type III functions.
3. The transcriptional profile of the Type II cells is less of a direct parallel to Type II cells in mice, but the authors acknowledge this and call them Type II-like before reverting to Type II, “for simplicity”. The existence of a different subset of gd T cells in the human thymus that is not strictly a parallel of mouse subsets should be considered and discussed.
4. In the discussion, the authors say that their results “indicate” that the activation of distinct gd T cell effector programs in the periphery is modulated by different NKR ligands. They do not show this in the paper, or provide references, so should use the word “suggest” or else provide evidence.

Reviewer #2 (Remarks to the Author):

In this important study, the authors performed single-cell transcriptome & TCR sequencing on $\gamma\delta$ thymocytes from human fetal thymus to investigate a burning question in the field, i.e. whether and how specific gd TCR rearrangements drive the differentiation of gd T cells into innate effector T cells in the thymus. They revealed that fetal thymic $\gamma\delta$ T cells are primed to innate type-1, -2like, or -3 phenotypes. This conclusion mutually supports previous studies like Tan et al, 2021, *Science Immunology*; Tieppo et al, 2019, *The Journal of experimental medicine*; Dimova et al, 2015, *PNAS*. Moreover, this study adds new interesting details about the differentiation of innate $\gamma\delta$ T effectors: The distinct effector lineages are not only committed in fetal thymus, but also prefer different public TCRs. This preference can be observed from the both samples analyzed, thus the authors argue that TCR-specific signals guide the commitment.

Furthermore, the authors report that innate effector $\gamma\delta$ thymocytes consist of both V γ 9V δ 2 TCR and non-V γ 9V δ 2 TCR, which was rarely reported before (or only by themselves). In sum, this study expands our understanding of the ontogeny of human innate $\gamma\delta$ T cells and is thus of general interest in the $\gamma\delta$ field.

Nonetheless, a few issues should be addressed before publication of this study.

Major:

1) The small sample size (2 samples with 4412 cells in total) and lack of validation is problematic. It is understandable that samples from human fetus are not easily collected. However, the sample size of 2 is much fewer than previous studies from the same group (Tieppo et al, 2019) and makes it difficult to discriminate donor specific factors and batch effects from gestation time specific factors (for example Figure 3). There is not FACS data to support the heterogeneity revealed by scRNAseq. The validation by bulk RNA seq in this study was also not based on type1, 2, 3 immunity, thus provided limited support to the major conclusions.

Along this line, it would be nice to have quasi-longitudinal data to address the role of the gd TCR in instructing (innate) gd T cell effector lineages in the fetal and / or neonatal thymus. Considering that the study only includes two samples from 14w and 21w, suggesting the difference between these two samples "was associated with the gestation time" probably requires more validation.

Other options might be to provide in vitro differentiation data or to tune down the conclusion that specific TCR sequences drive the differentiation into specific effector phenotypes. It is not clear from the current data that timing, or functional waves of fetal gd T cell development would not play a role here. The authors ignore that type1 / type3 effector gd T cells have been described to as early as in week 8 fetal thymus (ref 121).

Also, the wouldn't the observation that fetal V γ 9/V δ 2 and non V γ 9/V δ 2 TCR show very overlapping gene expression profiles argue somewhat against the hypothesis that TCR instruct gd T cell effector phenotypes (or against a "...role of the CDR3 in the maturation towards effector fates in the human fetal thymus.")?

2) In Fig. 6, the authors summarized by trajectory analysis two distinct differentiation trajectories for type1 and type2&3 innate $\gamma\delta$ T cells. However, the accuracy of trajectory analysis based on snapshot of gene expression is somewhat questionable. Absence of type1,2,3 lineage markers (RORC, MAF, TBET, CCR4, CCR6, etc) observed in Fig 6B further supports this question. To make this part of analysis more convincing, authors may consider: 1) to track cell differentiation trajectory using TCR repertoire as 'finger print'. For example, the type2 and type3 cells should share much of their TCR repertoires with their common progenitor, but not the progenitor of type1 cells. 2) to run trajectory analysis based on RNA splicing dynamics, for example scVelo or RNA Velocity. 3) to use the same UMAP and clustering as shown in Fig 1A to reduce confusion.

Minor:

a) Fig 1: The description for Fig 1 could be more brief and perhaps clearer, it will be easier for readers if clusters are introduced by the order of numbering. Also, trying to define a cluster by a couple of genes is not always convincing.

b) Fig 1C and all following heatmaps: the heatmaps by cluster lost the information of detection rate of selected DEGs, which is a special (dis)advantage of scRNAseq. Heatmap by each single-cell, or dot

plot that using size of dot to indicate detection rate will likely be more appreciated.

c) Fig 2E: why Th17-related pathways are solely enriched in type2 cells?

d) Figure S3B,C : text is too small to read.

e) Why did the authors suggest that the results in fig2E and FigS3B are similar? What are the 'well-known players' in Fig S3E? for example, RORC was not found in type3 cluster, GATA3 was not found in type2 cluster, TBX21 in type1 cluster seems lower than threshold.

f) Fig 4E call-out on page 10 – something may be mixed up?

g) Fig 4C: comparison to ab T cells appears a bit hard to interpret “at first look”, but I am not sure how to show the overlapping gene expression. very minor detail: SCART1 is a pseudogene

h) Fig 4D related description: It would be clearer to read if authors can specify in text that it is the analysis of TRDV2 CDR3 features.

i) Fig 5: this figure suggests that type1-3 $\gamma\delta$ thymocytes have different preference for V γ 9V δ 2 TCRs. To augment this point, a comparison of TCR repertoire of type 1, 2, 3 thymocytes is necessary.

j) Fig 5F, is the phenomenon that public V δ 2 chains have different pairing preference to public V γ 8 or V γ 9 chains, observed in the both samples?

k) In suppl. table 2, the authors refer to Rath et al 2020, but do not cite the study in their reference list. Instead, they republish that list genes of CTL-related genes. This might be considered as bad style.

Reviewer #3 (Remarks to the Author):

Sanchez and colleagues used single cell RNA sequencing and single cell $\gamma\delta$ T cell receptor sequencing on fetal and pediatric $\gamma\delta$ thymocytes to report on the development of human $\gamma\delta$ T cells. The authors assessed developmental stage-specific thymocyte effector clusters and their concomitant TCR repertoire. The authors demonstrate heterogeneity of immature and mature $\gamma\delta$ human fetal thymocytes using and identify 14 clusters. However, these clusters exhibit overlap in the expression of effector genes and the function of cluster 10 was not identified. The authors should explain why they chose to separate data into 14. In the case of pediatric thymocytes, the authors should explain why they identified 11 clusters as several key markers are shared between clusters. The authors should also examine whether $\gamma\delta$ thymocytes can also be separated into distinct populations by expression of key markers at the protein level. Markers that were used to determine the egress score should be listed. Authors should identify what proportion of cells in each cluster is in each cell cycle stage.

Reviewer #4 (Remarks to the Author):

Sanchez et al. apply scRNA-seq and scTCR-seq to study $\gamma\delta$ T cell development in fetal and pediatric thymi. They identify mature fetal $\gamma\delta$ T cells are committed to a type 1, 2, or 3-like effector fate which are associated with different CD3R sequences. They identify pediatric thymus is heterogeneous, containing type1/type 3 effector profiles. The novelty of this study is its analysis in human tissues to better decipher the developmental trajectories of $\gamma\delta$ T cells in humans. This study contributes these original findings to the field and addresses the limitations of studies performed in mice. Such studies in mice cannot be directly translated to humans due to the lack of γ and δ loci conservation.

Methodology is appropriate for this study. Sufficient detail in methods is provided for reproducibility. While the challenge of acquiring human thymi samples is recognized, the data may be more robust with greater than n=2 fetuses and n=2 pediatric patient sample numbers. Reviewer appreciates validation of scRNA-seq findings by bulk RNA-seq on sorted cell populations. However, there are no validation measures applied to detect protein expression of effector subsets (for ex. Flow cytometry, CyTOF, Western blot) in fetal or pediatric subsets, which would strengthen authors' conclusions.

Reviewer recommends revisions at this stage.

Specific questions and comments:

1. Fig 2 illustrates mature thymocytes are committed to type 1, 2, 3 effector states, which is intriguing. It can be better clarified in the discussion why this may be and what may contribute to the programming of mature fetal thymocytes to be predisposed towards either type 1, 2-like, or 3 effector states. It is discussed that “the CDR3 sequence contributes to a difference in TCR signal strength during maturation and thus to the type of effector fate.”, on p15. To strengthen this argument, reviewer suggests to experimentally test this and/or provide functional validation for supporting evidence of the three developmental pathways identified.

2. Downregulation of RUNX1 in Fig. 6 is suggested to “correspond with the timing of early and strong TCR signaling”.. and seen in mouse CD4 $\alpha\beta$ T cells. What is the role of RUNX1 in mature $\gamma\delta$ T cells? Runx1-bound enhancer elements have been shown to change $\gamma\delta$ T cell lineage dependent expression of PLZF. See Mao et al, Nature Communications, 2017 Oct 16;8(1):863. Is a similar phenomenon observed here?

3. p. 11, line 27. Discussion on cluster 9 enriched genes. “These features indicate that this cluster 9 can be the source of the anti-cancer Vdelta1+NKp30+ gamma delta T cells in the blood adult subjects.” It’s recommended that this statement be elaborated on; as it stands, appears as an over-interpretation of the data.

4. What are the overall implications of the human pediatric thymus exhibiting combined features of type 1 and type 3 immunity compared to the fetal thymus “programming” of type 1, 2-like, and 3 with regards to aging? An overall view of the findings of distinct trajectories in fetal vs. pediatric, different effector subsets identified in each, and overall implications and significance of these can be better discussed.

Point-by-point response to the reviewers' comments

Reviewer #1

In this manuscript, the authors present a comprehensive analysis of gamma-delta T cell subsets in human fetal and postnatal thymocytes using scRNAseq and TCR sequencing. While mouse gd T cell development has been well-studied, much less is known about gd T cell development in humans. Comparisons of gd T cells between mice and humans have been complicated by the lack of direct homology between T cell receptor chains. However, the underlying regulators of effector function appear to be mostly conserved, allowing more useful comparisons by transcriptomics studies.

Here, the authors have asked the question of whether TCR identity is linked to effector function during development in the human thymus. To answer this question, they defined multiple subsets (cluster) of gd T cells in fetal and pediatric human thymus by single cell transcriptome and TCR analysis. This approach built upon earlier studies by this group and others using bulk RNAseq and FACS, and provided a much more in-depth analysis of the molecular signatures carried by different cells regarding of their TCR chain identities.

Surprisingly, different TCRg and TCRd chains were not correlated with effector function, as they are in mice. Instead, gd T cell subsets expressing any combination of TCRg and TCRd chains could adopt any of the three major functional programs. However, certain "public" CDR3 sequences were strongly correlated with effector function. This included the Vg9Vd2 subset, which has been a subset of some contention. These results are consistent with an

instructive role for TCR signal strength in functional programming during development, as occurs in mice, but with more flexible gdTCR chain requirements.

Importantly, the transcriptional signatures of gd T cell subsets in the fetal thymus suggest a developmental trajectory that is consistent with that seen in fetal mouse gd T cell development. The results of this study indicate an early strong TCR signal for Type I effector programming, followed by differential TCR signaling in Type II and Type III gd T cells in the more mature stages of development. These critical findings validate the relevance of studies of mouse gd T cell development and function, and eliminates concerns about the lack of conservation between mouse and human TCR chains in identifying effector function. It also clears the way for combining the power of mechanistic studies in mice with detailed molecular analysis in humans, in an iterative manner. Together, these results are a significant advance in the field, and should have a major impact on future studies of the development and function of gd T cells in both humans and mice.

We thank the reviewer for this nice summary and highlighting the important points of our study.

Reviewer # 1, major points

Reviewer # 1, point 1

A more comprehensive discussion of the recently reported paper in Science Immunology by Prinz and colleagues on human gd T cells is warranted given the similarities and differences of their findings in the thymus versus their findings in the periphery.

We followed this suggestion and have included a more comprehensive discussion regarding the similarities and differences between our study and the one by Prinz and colleagues (Tan et al 2021. Science Immunology) (highlighted in the Discussion of the revised manuscript, page 15)

Reviewer # 1, point 2

The identification of Type I and Type III gd T cell subsets are well supported. Indeed, the finding of IL-17A and IL-17F in the Type III subset in the fetal thymus supports a previous study that defined Type III gd T cells in cord blood. These cells which did not express IL-17 at the mRNA level (Prinz and colleagues), suggesting that in the periphery, stimulation is needed for these cells to elaborate their Type III functions.

We discuss this point now in more detail in the discussion of the revised manuscript (page 15, see also reviewer # 1, point 1). Furthermore, we have included now also new data regarding the evolution of the type 3 $\gamma\delta$ thymocyte subset during gestation (new Figure 3E: percentage of FT type 3 (and type 1 and type 2) according to gestation age; new supplementary Figure 4A with type 3 score according to gestation time).

Reviewer # 1, point 3

The transcriptional profile of the Type II cells is less of a direct parallel to Type II cells in mice, but the authors acknowledge this and call them Type II-like before reverting to Type II, "for simplicity". The existence of a different subset of gd T cells in the human thymus that is not strictly a parallel of mouse subsets should be considered and discussed.

We appreciate this suggestion of the reviewer. We discuss now the parallels and differences between the human fetal type 2 thymocytes and the mouse $\gamma\delta$ T cell subset shown to produce readily IL4, also known as iNKT-like $\gamma\delta$ cells or $\gamma\delta$ NKT cells, that are enriched for the expression of the V γ 1 and V δ 6 chains (Discussion, page 14 of the revised manuscript).

Reviewer # 1, point 4

In the discussion, the authors say that their results "indicate" that the activation of distinct gd T cell effector programs in the periphery is modulated by different NKR ligands. They do not show this in the paper, or provide references, so should use the word "suggest" or else provide evidence.

We use now the work 'suggest'.

In addition, we provide now also confirmation of differential NKR expression between the distinct fetal $\gamma\delta$ effector thymocytes at the protein level by flow cytometry (new Figure 3 F and new Supplemental Figure 5B) and have adapted the results (page 12 revised manuscript) and discussion (page 14 revised manuscript) sections regarding NKR expression.

Reviewer #2

In this important study, the authors performed single-cell transcriptome & TCR sequencing on gd thymocytes from human fetal thymus to investigate a burning question in the field, i.e. whether and how specific gd TCR rearrangements drive the differentiation of gd T cells into innate effector T cells in the thymus. They revealed that fetal thymic gdT cells are primed to innate type-1, -2like, or -3 phenotypes. This conclusion mutually supports previous studies like Tan et al, 2021, Science Immunology; Tieppo et al, 2019, The Journal of experimental medicine; Dimova et al, 2015, PNAS. Moreover, this study adds new interesting details about the differentiation of innate gdT effectors: The distinct effector lineages are not only committed in fetal thymus, but also prefer different public TCRs. This preference can be observed from the both samples analyzed, thus the authors argue that TCR-specific signals guide the commitment.

Furthermore, the authors report that innate effector gd thymocytes consist of both V γ 9V δ 2 TCR and non-V γ 9V δ 2 TCR, which was rarely reported before (or only by themselves). In sum, this study expands our understanding of the ontogeny of human innate gdT cells and is thus of general interest in the gd field.

We thank the reviewer for this nice summary and highlighting the important points of our study.

Reviewer # 2, major point 1

The small sample size (2 samples with 4412 cells in total) and lack of validation is problematic. It is understandable that samples from human fetus are not easily collected. However, the sample size of 2 is much fewer than previous studies from the same group (Tieppo et al, 2019) and makes it difficult to discriminate donor specific factors and batch effects from gestation time specific factors (for example Figure 3). There is not FACS data to support the heterogeneity revealed by scRNAseq. The validation by bulk RNA seq in this study was also not based on type1, 2, 3 immunity, thus provided limited support to the major conclusions. Along this line, it would be nice to have quasi-longitudinal data to address the role of the gd TCR in instructing (innate) gd T cell effector lineages in the fetal and / or neonatal thymus. Considering that the study only includes two samples from 14w and 21w, suggesting the difference between these two samples "was associated with the gestation time" probably requires more validation.

We were able to increase the fetal thymus (FT) group with 4 more fetuses, thus increasing till 6 fetal samples with 16,508 cells in total to be analyzed by our scRNA+TCR approach. The gestation time ranges from 14w till 22w (14w0d; 15w2d; 16w2d; 17w5d; 21w0d; 22w6d).

As such, we do have now a quasi-longitudinal series of data according to gestation time.

Thus we were able to reveal novel associations: the type 3 effector program decreased during gestation (new Figure 3E and new supplemental Figure 4) while the percentage of type 1 effector increased (new Figure 3E), thus showing wave-like patterns. We found also a clear increase in TdT expression during gestation (new Supplementary Figure 7B) that was associated with an increase in N addition (new Supplementary Figure 7A).

Furthermore, we performed flow cytometry confirming a series of findings at the protein level (n= 9), that are now integrated in our figures, including the division in type 1, type 3 and type 2-like effector clusters (for example see new main Figure panel 3F and new Supplementary Figure 5B).

Other options might be to provide in vitro differentiation data or to tune down the conclusion that specific TCR sequences drive the differentiation into specific effector phenotypes. It is not clear from the current data that timing, or functional waves of fetal gd T cell development would not play a role here. The authors ignore that type1 / type3 effector gd T cells have been described to as early as in week 8 fetal thymus (ref 121).

Our new data confirm the observation that there is no association between the number of N additions in the CDR3 repertoire and the functional effector program. As the number of N additions is clearly associated with the gestation time (new Supplementary Figure 7A), this observation argues against a different timing in the generation of particular CDR3 sequences (as observed in the mouse model) as a possible explanation for their association with

particular effector fates. As detailed under Reviewer # 1, point 1, we included a more comprehensive discussion regarding the similarities and differences between our study and the one by Prinz and colleagues (Tan et al 2021. Science Immunology) (highlighted in the Discussion of the revised manuscript, page 15), especially regarding the different observations at particular gestation times.

Also, the wouldn't the observation that fetal Vg9/Vd2 and non Vg9/Vd2 TCR show very overlapping gene expression profiles argue somewhat against the hypothesis that TCR instruct gd T cell effector phenotypes (or against a "...role of the CDR3 in the maturation towards effector fates in the human fetal thymus.")?

We understand this remark, but we do think that the observation of similar gene expression profiles in both V γ 9V δ 2 and nonV γ 9V δ 2 T cells is not necessarily in contradiction with an involvement of TCR signaling in instructing the effector phenotype. Indeed, it appears that especially the CDR3 of the TRDV2-containing δ chain is associated with the effector program (new Figure 6) rather than the public TRGV9-containing CALWEVQELGKKIKVF or TRGV8-containing CATWDTTGWFKIF CDR3 sequence (new supplemental Figure 9B-C). Despite that our new CDR3 data, obtained from the 6 different human fetuses, confirms the association between particular TRDV2-containing CDR3 sequences and the $\gamma\delta$ effector phenotype, it also shows high variability indicating that other signals, besides the one of $\gamma\delta$ TCR, are also instructing the $\gamma\delta$ thymocyte effector phenotype. We have included this notion in the revised version of the manuscript (highlighted in the Discussion, page 16).

Note that we confirmed as well the prevalence of both V γ 9V δ 2 and nonV γ 9V δ 2 T cells in the three effector types at the protein level (new Figure 5D, right panel).

Reviewer # 2, major point 2

2a.

In Fig. 6, the authors summarized by trajectory analysis two distinct differentiation trajectories for type1 and type2&3 innate gdT cells. However, the accuracy of trajectory analysis based on snapshot of gene expression is somewhat questionable. Absence of type1,2,3 lineage markers (RORC, MAF, TBET, CCR4, CCR6, etc) observed in Fig 6B further supports this question.

We apologize to the reviewers for the potential confusion generated with this part as the old UMAP figure overlaying the lineage trajectories was different than the general one present in the old Figure 1. We have included now (new Supplementary Figure 10) a series of type 1, type 3 and type 2 effector cluster markers in the new UMAP plots used for the lineage tracing. Note that these effector cluster markers have not been included in the main figure (Figure 7B) as the goal of that figure is to show differential expressed markers in the type 1 lineage compared to the others across the differentiation pathway rather than at the endpoint. This has now also been clarified in the legend of the new Figure 7.

2b.

To make this part of analysis more convincing, authors may consider: 1) to track cell differentiation trajectory using TCR repertoire as 'finger print'. For example, the type2 and type3 cells should share much of their TCR repertoires with their common progenitor, but not the progenitor of type1 cells. 2) to run trajectory analysis based on RNA splicing dynamics, for example scVelo or RNA Velocity. 3) to use the same UMAP and clustering as shown in Fig 1A to reduce confusion.

We thank the reviewer for these suggestions. We followed the first suggestion and, interestingly, observed an enrichment of a specific TRDV2 sequence (CACDTGGYSWDTRQMFF) across the whole lineage described for type 1 cells (new Figure

7C). This result is thus in line with a specific developmental pathway for the type 1 effector program as suggested by the trajectory analysis obtained with the Slingshot package. Note that the Slingshot software can only compute trajectories with a single root. As a consequence, the three different lineages are starting from the same point and therefore no differences at the level of CDR3 sequences of progenitor cells can be observed. We included now a more in-depth explanation in the methods section (highlighted in the methods section, page 22) to clarify why the UMAP displayed in new Figure 7A is not the same as in new Figure 2A.

Reviewer # 2, minor points

Reviewer # 2, minor point a:

Fig 1: The description for Fig 1 could be more brief and perhaps clearer, it will be easier for readers if clusters are introduced by the order of numbering. Also, trying to define a cluster by a couple of genes is not always convincing.

We have followed the suggestion of the reviewer and shortened the text and tried to make it more clear.

Reviewer # 2, minor point b:

Fig 1C and all following heatmaps: the heatmaps by cluster lost the information of detection rate of selected DEGs, which is a special (dis)advantage of scRNAseq. Heatmap by each single-cell, or dot plot that using size of dot to indicate detection rate will likely be more appreciated.

We thank the reviewer for the suggestion. We updated the heatmaps to reflect the expression of markers at single-cell level in the distinct clusters (new Figure 2C, new Supplementary Figure 11).

Reviewer # 2, minor point c:

Fig 2E: why Th17-related pathways are solely enriched in type2 cells?

We apologize to the reviewer for this mistake: in the original submission we switched the order of the column identifiers in this figure panel. This mistake now is corrected (new supplemental Figure 3B)

Reviewer # 2, minor point d:

Figure S3B,C : text is too small to read.

We agree with the suggestion of the reviewer. However, we decided to remove this figure as the main points of this panel are already illustrated in new Figure 3B-C, new Supplementary Figure 3B and new Supplementary Figure 14, that is: to highlight that the differential expressed genes in type 1/3 could be linked to physiological roles.

Reviewer # 2, minor point e:

Why did the authors suggest that the results in fig2E and FigS3B are similar? What are the 'well-known players' in Fig S3E? for example, RORC was not found in type3 cluster, GATA3 was not found in type2 cluster, TBX21 in type1 cluster seems lower than threshold.

We apologize to the reviewer for the confusion generated with the old FigS3C. We agree that we did not provide enough explanation regarding the "well-known players" sentence. Due to this reason and the reason described in the answer to minor point d above we decided to delete this figure.

Reviewer # 2, minor point f:

Fig 4E call-out on page 10 – something may be mixed up?

Indeed, there was a mistake and we apologize for it. This has now been corrected in the new numbering of the figures.

Reviewer # 2, minor point g:

Fig 4C: comparison to ab T cells appears a bit hard to interpret “at first look”, but I am not sure how to show the overlapping gene expression.

We have now adapted this figure: we moved the comparisons with $\alpha\beta$ T thymocytes (bulk RNAseq) to Supplementary Figure 8A while we included now the protein data on V γ 9V δ 2 vs nonV γ 9V δ 2 in the type 1/3/2 effector groups in the new Figure 5D (right panel).

very minor detail: SCART1 is a pseudogene

We thank the reviewer for this remark. Because of this we decided to remove the *SCART1* mentions and figures from the manuscript.

Reviewer # 2, minor point h:

Fig 4D related description: It would be clearer to read if authors can specify in text that it is the analysis of TRDV2 CDR3 features.

We have specified this now in the text ('Analysis... among TRDV2-containing CDR3 sequences...': results section, page 9 of the revised manuscript).

Reviewer # 2, minor point i:

Fig 5: this figure suggests that type1-3 gd thymocytes have different preference for V γ 9V δ 2 TCRs.

The old Figure 4B (that the reviewer is probably referring to) is describing the abundance of cells with paired V γ 9 and V δ 2 chains in each effector cluster at transcriptomic level, thus this figure is not describing any specific V γ 9V δ 2 TCR abundance. This figure has been updated (new Figure 5D) and also includes now protein data that highlights how V γ 9V δ 2 and nonV γ 9V δ 2 fetal $\gamma\delta$ thymocytes display similar effector programming.

Reviewer # 2, minor point j:

Fig 5F, is the phenomenon that public V δ 2 chains have different pairing preference to public V γ 8 or V γ 9 chains, observed in the both samples.

Yes, this is indeed the case. Now we have updated this figure with data from the four new fetal thymuses (n=6 in total) in new Figure 6C and performed a statistical analysis.

Reviewer # 2, minor point k:

In suppl. table 2, the authors refer to Rath et al 2020, but do not cite the study in their reference list. Instead, they republish that list genes of CTL-related genes. This might be considered as bad style.

It was not our intention not to include Ra et al in the reference list. Besides in the heading of supplemental Table 2 (now Supplemental Table 5), this reference is now included in the reference list and we apologize to have omitted this reference from the reference list before.

Reviewer #3

Reviewer #3, point 1

Sanchez and colleagues used single cell RNA sequencing and single cell gd T cell receptor sequencing on fetal and pediatric gd thymocytes to report on the development of human gd T cells. The authors assessed developmental stage-specific thymocyte effector clusters and their concomitant TCR repertoire. The authors demonstrate heterogeneity of immature and mature gd human fetal thymocytes using and identify 14 clusters. However, these clusters exhibit overlap in the expression of effector genes and the function of cluster 10 was not identified. The authors should explain why they chose to separate data into 14. In the case of pediatric thymocytes, the authors should explain why they identified 11 clusters as several key markers are shared between clusters.

We thank the reviewer for highlighting the key points of our manuscript. Both the fetal and pediatric UMAP figures have been updated now after including four new fetal thymus and one new pediatric thymus. Thus the number of clusters are not the same as before. We understand the comment of the reviewer as in the methods section we included the details to obtain the same number of clusters with our dataset (clustering resolution algorithm), but we did not describe the exact process to decide why we used a specific resolution of the clustering algorithm to obtain the results. In brief, we assessed the clusters generated after using different resolutions of the clustering algorithm and we compared their transcriptomic profile with the other clusters to check that we were not “falling short” in the number of clusters or over-clustering our data. Other R packages, such as *ClusterTree*, use a similar system as ours and also conclude their pipeline with the step of examination by the user of the clusters generated at different resolutions. We include now a short statement in methods section to clarify this. Note that a certain level of similarity between clusters is expected as our dataset is composed of a homogenous population ($\gamma\delta$ +CD3+ thymocytes) that are undergoing a process of maturation towards a mature cell state.

Reviewer #3, point 2

The authors should also examine whether gd thymocytes can also be separated into distinct populations by expression of key markers at the protein level.

We performed flow cytometry confirming a series of findings at the protein level, that are now integrated in our figures, including the separation into type 1, type 3 and type 2-like effector clusters (for example see new Figure 3F and new Supplementary Figure 5).

Reviewer #3, point 3

Markers that were used to determine the egress score should be listed.

The markers were listed in Supplementary Table 2 but this was not mentioned in the main text. We apologize for this error. We are now referring to the Supplementary Table 2 when we introduce the egress score in the main text.

Reviewer #3, point 4.

Authors should identify what proportion of cells in each cluster is in each cell cycle stage.

We agree with the comment of the reviewer as the previous figure connected to this statement (old Supplementary Figure 1C and Supplementary Figure 1E) did not allow the quantification of the different cycling states in each cell cluster. We have generated new figures that represent this more clearly (Supplementary Figure 2C and Supplementary Figure 11E).

Reviewer #4

Sanchez et al. apply scRNA-seq and scTCR-seq to study gd T cell development in fetal and pediatric thymi. They identify mature fetal gd T cells are committed to a type 1, 2, or 3-like effector fate which are associated with different CDR3 sequences. They identify pediatric thymus is heterogeneous, containing type1/type 3 effector profiles. The novelty of this study is its analysis in human tissues to better decipher the developmental trajectories of gd T cells in humans. This study contributes these original findings to the field and addresses the limitations of studies performed in mice. Such studies in mice cannot be directly translated to humans due to the lack of γ and δ loci conservation.

We thank the reviewer for highlighting these important points.

Reviewer #4, major comment

While the challenge of acquiring human thymi samples is recognized, the data may be more robust with greater than n=2 fetuses and n=2 pediatric patient sample numbers. Reviewer appreciates validation of scRNA-seq findings by bulk RNA-seq on sorted cell populations. However, there are no validation measures applied to detect protein expression of effector subsets (for ex. Flow cytometry, CyTOF, Western blot) in fetal or pediatric subsets, which would strengthen authors' conclusions.

We were able to increase the fetal thymus group with 4 more fetuses, thus increasing till 6 fetal samples with 16,508 cells in total to be analyzed by our scRNA+TCR approach. The gestation time ranges from 14w till 22w (14w0d; 15w2d; 16w2d; 17w5d; 21w0d; 22w6d). As such, we do have now a quasi-longitudinal series of data according to gestation time. Thus we were able to reveal novel associations: the type 3 effector program decreased during gestation (new Figure 3E and new Supplementary Figure 4) while the percentage of type 1 effector increased (new Figure 3E), thus showing wave-like patterns. We found also a clear increase in TdT expression during gestation that was associated with an increase in N addition (new Supplementary Figure 7).

Furthermore, we performed flow cytometry confirming a series of findings at the protein level (n=9 for the fetal thymus group), that are now integrated in our figures, including the division in of type 1, type 3 and type 2-like effector clusters (for example see new Figure 3F). The pediatric thymus group was increased as well with one more sample (11.0 years old) for the scRNA+TCR approach and we validated also here main findings at the protein level (n=8) (see for example new Figure 8 F-G).

Reviewer #4, point 1

Fig 2 illustrates mature thymocytes are committed to type 1, 2, 3 effector states, which is intriguing. It can be better clarified in the discussion why this may be and what may contribute to the programming of mature fetal thymocytes to be predisposed towards either type 1, 2-like, or 3 effector states. It is discussed that "the CDR3 sequence contributes to a difference in TCR signal strength during maturation and thus to the type of effector fate.", on p15. To strengthen this argument, reviewer suggests to experimentally test this and/or provide functional validation for supporting evidence of the three developmental pathways identified.

We think that the functional validation of the influence of the CDR3/TCR signaling versus other possible mechanisms is outside the scope of the present study. However, as detailed under 'Reviewer #4, major comment' we have validated the division into type 1, 2 and 3 effector fates at the protein level by flow cytometry. Furthermore, having now scGEX+TCR data on 6 different fetuses along a range of gestation time (14w-22), we provide new data on the link of CDR3 data with the trajectory analysis (new main Figure 8C and D) and data on the potential precursor of the type 3 effector program (new Supplementary Figure 4B) to indicate that both CDR3-dependent and -independent mechanisms could be involved.

This is now also now better clarified in the Discussion section (page 16 of the revised manuscript):

‘...propose that the CDR3 sequence contributes to a difference in TCR signal strength during maturation and thus, together with signaling from other receptor types such as NKR and/or cytokine receptors and/or precursor frequencies^{59,74,75}, to the type of effector fate.’

Reviewer #4, point 2

Downregulation of RUNX1 in Fig. 6 is suggested to “correspond with the timing of early and strong TCR signaling”.. and seen in mouse CD4 ab T cells. What is the role of RUNX1 in mature gd T cells? Runx1-bound enhancer elements have been shown to change gd T cell lineage dependent expression of PLZF. See Mao et al, Nature Communications, 2017 Oct 16;8(1):863. Is a similar phenomenon observed here?

We thank the reviewer for raising this interesting point. In this paper Mao and colleagues identified a critical enhancer, enriched for canonical motifs for the TF Runx1, that controls PLZF expression exclusively in innate lymphoid lineages. The enrichment of *RUNX1* expression across the T3 and T2 lineages (new Supplementary Figure 10) goes in line with the observations described by Mao et al as Type 3 and Type 2-like cluster displayed higher levels of *PLZF* expression compared with the other mature effector cluster (Type 1). Due to this we decided to describe these expression patterns of *PLZF* and *Runx1* in the main text referring to Mao et al. (Results section, page 11).

Reviewer #4, point 3

p. 11, line 27. Discussion on cluster 9 enriched genes. “These features indicate that this cluster 9 can be the source of the anti-cancer Vdelta1+Nkp30+ gamma delta T cells in the blood adult subjects.” It’s recommended that this statement be elaborated on; as it stands, appears as an over-interpretation of the data.

We have now validated the high specific expression of Nkp30 on the Vδ1+ γδ pediatric thymocytes at the protein level (new Supplementary Figure 12D). Furthermore, we have rewritten this part in order to make the interpretation of the data more clear (Results section, page 12 of the revised manuscript).

Reviewer #4, point 4

What are the overall implications of the human pediatric thymus exhibiting combined features of type 1 and type 3 immunity compared to the fetal thymus “programming” of type 1, 2-like, and 3 with regards to aging? An overall view of the findings of distinct trajectories in fetal vs. pediatric, different effector subsets identified in each, and overall implications and significance of these can be better discussed.

We have added now overall possible implications of our finding in the Discussion (highlighted on page 15 of the revised manuscript).

REVIEWERS' COMMENTS

Reviewer #1 (Remarks to the Author):

The authors have done a great job of addressing my questions and comments. No further revisions are needed.

Reviewer #2 (Remarks to the Author):

The authors have done a great job replying to all reviewers' points and revising the ms, including increasing sample sizes.

The ms should be accepted for publication.

Reviewer #4 (Remarks to the Author):

Sanchez et al. have sufficiently addressed the reviewers' concerns in their revised manuscript. The increase of sample size particularly (ie. 6 fetal thymi) has strengthened the data analysis compared to the original submission. This study overall contributes to the field, advancing the current knowledge on gamma-delta T cell development in the human thymus. No further issues are to be addressed.

Response to the reviewers' comments

Reviewer #1 (Remarks to the Author):

The authors have done a great job of addressing my questions and comments. No further revisions are needed.

Reviewer #2 (Remarks to the Author):

The authors have done a great job replying to all reviewers' points and revising the ms, including increasing sample sizes.

The ms should be accepted for publication.

Reviewer #4 (Remarks to the Author):

Sanchez et al. have sufficiently addressed the reviewers' concerns in their revised manuscript. The increase of sample size particularly (ie. 6 fetal thymi) has strengthened the data analysis compared to the original submission. This study overall contributes to the field, advancing the current knowledge on gamma-delta T cell development in the human thymus. No further issues are to be addressed.

Our response:

We thank the reviewers for these very positive comments.